# Biologically Constrained Barrel Cortex Model Integrates Whisker Inputs and Replicates Key Brain Network Dynamics

**Tianfang Zhu**[1*]**, Dongli Hu**[1*]**, Jiandong Zhou**[1]**, Kai Du**[2,4†] **& Anan Li** [1,3,5†]

[1]Wuhan National Laboratory for Optoelectronics, Huazhong University of Science and Technology
[2]Institute for Artificial Intelligence, Peking University
[3]HUST-Suzhou Institute for Brainsmatics, JITRI
[4]State Key Laboratory of General Artificial Intelligence, Peking University
[5]School of Biomedical Engineering, Hainan University
[1]{funfunfun,hudongli,jiandongzhou,aali}@hust.edu.cn
[2]{kai.du}@pku.edu.cn

## Abstract

The brain's ability to transform sensory inputs into motor functions is central to neuroscience and crucial for the development of embodied intelligence. Sensory-motor integration involves complex neural circuits, diverse neuronal types, and intricate intercellular connections. Bridging the gap between biological realism and behavioral functionality presents a formidable challenge. In this study, we focus on the columnar structure of the superficial layers of mouse barrel cortex as a model system. We constructed a model comprising 4,218 neurons across 13 neuronal subtypes, with neural distribution and connection strengths constrained by anatomical experimental findings. A key innovation of our work is the development of an effective construction and training pipeline tailored for this biologically constrained model. Additionally, we converted an existing simulated whisker sweep dataset into a spiking-based format, enabling our network to be trained and tested on neural signals that more closely mimic those observed in biological systems. The results of object discrimination utilizing whisker signals demonstrate that our barrel cortex model, grounded in biological constraints, achieves a classification accuracy exceeds classical convolutional neural networks (CNNs), recurrent neural networks (RNNs), and long short-term memory networks (LSTMs), by an average of 8.6%, and is on par with recent spiking neural networks (SNNs) in performance. Interestingly, a whisker deprivation experiment, designed in accordance with neuroscience practices, further validates the perceptual capabilities of our model in behavioral tasks. Critically, it offers significant biological interpretability: post-training analysis reveals that neurons within our model exhibit firing characteristics and distribution patterns similar to those observed in the actual neuronal systems of the barrel cortex. This study advances our understanding of neural processing in the barrel cortex and exemplifies how integrating detailed biological structures into neural network models can enhance both scientific inquiry and artificial intelligence applications. The code is available at https://github.com/fun0515/RSNN_bfd.

## 1 Introduction

The brain's remarkable ability to transform sensory inputs into coordinated motor actions is pivotal, not only in neuroscience but also in the development of embodied intelligence systems (Bartolozzi et al., 2022; Putra et al., 2024). This process, known as sensory-motor integration, relies on complex neural circuits, diverse neuronal types, and intricate inter-neuronal connections (Karadimas et al., 2020; Muñoz-Castañeda et al., 2021; Ziegler et al., 2023). These elements are fundamental for both

---

*Equal contribution. [†] Corresponding authors.

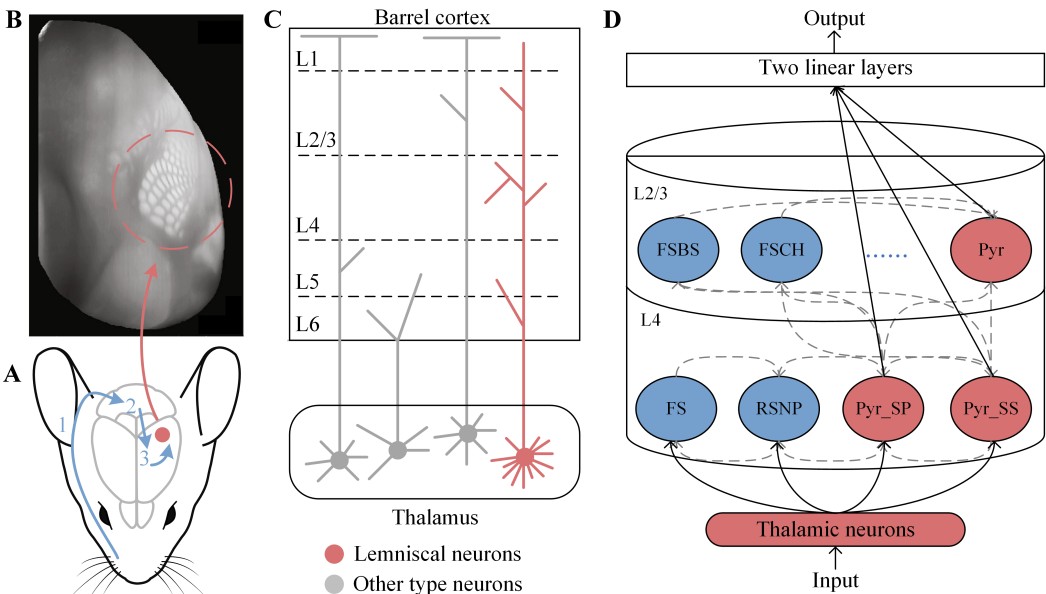

Figure 1: **The biophysical modeling of the barrel cortex based on real anatomy.** (**A**) The "whisker-barrel" signaling pathway. Sensory signals from the whiskers are transmitted by the trigeminal ganglion neurons (1) to the brainstem neurons (2), then relayed to the barrel cortex via the thalamus neurons (3). (Figure modified from (Petersen, 2019)). (**B**) An anatomical image of the barrel cortex in adult mice, sourced from the Allen Brain Atlas (Wang et al., 2020). (**C**) Thalamic projection neurons directed to the barrel cortex. Those highlighted in red represent core thalamic projection neurons, originating from the ventral posteromedial thalamic nucleus, predominantly projecting to layer 4 (L4) of the cortex. (**D**) Outline of our biologically constrained network architecture. External inputs are conveyed into the network through 200 thalamic neurons. The network comprises 13 neuronal populations with a total of 4218 adaptive Leaky Integrate-and-Fire neurons. Red signifies excitatory neurons, whereas blue signifies inhibitory neurons. For instance, "Pyr_" denotes a specific subtype within the category of pyramidal neurons. The network's prediction is readout from the state of three excitatory populations within the network by two linear layers. Detailed information of 13 neuronal subtypes can be found in the supplementary material.

understanding brain function and advancing artificial intelligence systems capable of interacting seamlessly with the physical environment. However, achieving a balance between biological realism and functional performance in computational models remains a significant challenge, as these models must accurately replicate these complex biological processes.

The rodent barrel cortex, which processes tactile information from whiskers, serves as an exemplary model system for studying sensory-motor integration. Its well-documented columnar organization and neuronal circuitry provide a robust framework for investigations (Petersen, 2019; Staiger & Petersen, 2021). Despite this, traditional computational models often oversimplify biological complexities to achieve functionality, thus losing crucial insights into the actual neural mechanisms. Conversely, highly detailed biologically realistic models (Fan & Markram, 2019; D'Angelo & Jirsa, 2022), while rich in data, are often computationally intensive and difficult to train, limiting their practical application in simulating behaviorally relevant tasks.

To address these challenges, our study introduces an effective construction and training pipeline specifically designed for a biologically constrained model of barrel cortex, aimed at achieving bioplausible sensory-motor integration. Our approach is characterized by three primary contributions:

- **Biologically constrained barrel cortex model:** We developed a columnar model of the mouse barrel cortex, comprising 4,218 neurons across 13 neuronal subtypes. This model meticulously reflects the neural distribution and synaptic connections dictated by experimental anatomical findings (Huang et al., 2022), ensuring an authentic replication of the biological structure of mouse barrel cortex.

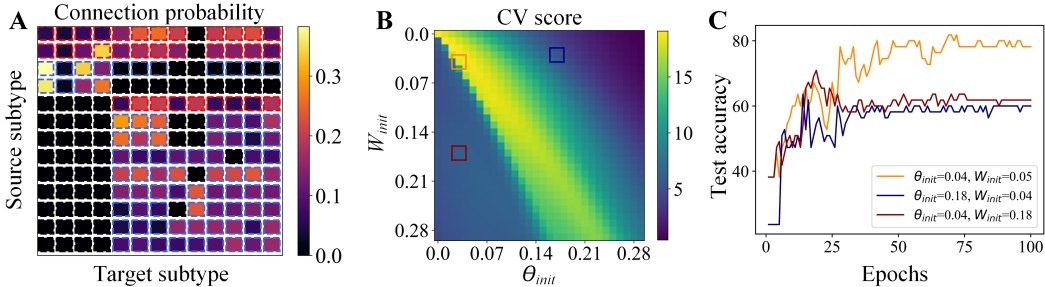

Figure 2: **Anatomical connection probability and network initialization.** (**A**) Connection probabilities among 13 neuron subtypes, derived from (Huang et al., 2022), with excitatory connections represented in red borders and inhibitory connections in blue. (**B**) The impact of two significant parameters $\theta_{init}$ and $W_{init}$ discussed in Sec. 3.2. Network's initial states are assessed using the coefficient of variation (CV) to measure the synchrony of neuronal firing. (**C**) Accuracy on a whisker-related dataset (see in Sec. 3.3) for three initial settings, as denoted by rectangle symbols in (**B**).

- **Pipeline for model construction and training:** Our novel pipeline is specifically tailored to balance the biological authenticity and behavioral functionality. This includes using intuitive neuron models and sparse connections to mimic the neural diversity and projection strength, introducing an order parameter to measure the synchrony of neuronal activity, thereby identifying trainable initial states. This pipeline is not only applicable to the barrel cortex but can also be extended to other biologically constrained models.

- **Establishing a novel spiking whisker sweep dataset:** We leveraged a previously developed simulated whisker sweep dataset (Zhuang et al., 2017), originally generated using a physical engine (Coumans & Bai, 2016–2021) with parameters informed by neuroscience research. Our significant contribution was transforming this simulated data into a novel spiking-based dataset (see in Fig. 3), which emulates the sensory signals from the thalamus area. This newly established spiking dataset provides a biologically meaningful platform for training and evaluating our model's sensory-motor integration capabilities.

While trained on the Spiking Whisker Sweep Dataset (see in Fig. 3), the classification accuracy of our model surpasses that of mainstream models like CNNs (Zhuang et al., 2017), RNNs (Collins et al., 2017), and LSTMs (Ullah et al., 2017), and is competitive with recent SNNs (Yin et al., 2020; Yu et al., 2022; Yao et al., 2021), particularly in neuroscience-inspired whisker deprivation experiments. Additionally, our model demonstrates superior biological interpretability. Our post-training analysis shows that neurons within our network exhibit firing patterns and distribution characteristics remarkably similar to those observed in an actual barrel cortex, including aspects such as response selectivity, dynamical gradients, and node degree distribution. Our findings underscore the potential of incorporating biological constraints into neural network models to enhance their performance on sensory-motor tasks and provide deeper insights into the underlying neural mechanisms. This approach promises significant advances in developing AI systems with brain-like perceptual and motor capabilities and contributes to neuroscientific efforts designed to unravel the complexities of brain function through advanced computational modeling.

## 2 RELATED WORK

**Barrel cortex:** The barrel cortex, a distinct area discovered within the primary somatosensory cortex of rodents, assumes a pivotal role in the animal's spatial perception, precise localization capabilities, and motor functions. Neurons within this cortex are organized into mesoscale barrel-like formations (see in Fig. 1B), aligned with the layout of the animal's snout whiskers, hence the name. Each "barrel" preferentially responds to tactile stimuli from the corresponding whisker, forming a distinctive "whisker-barrel" sensory pathway (Petersen, 2019; Staiger & Petersen, 2021). The general pathway process unfolds as follows: Sensory stimuli originating from the whiskers are initially conveyed through the brainstem, subsequently relayed to the barrel cortex via the thalamus, facilitating the integration and processing of tactile information (see in Fig. 1A). The unique structure and function of this "whisker-barrel" model make it a prevalent tool for studying information in-

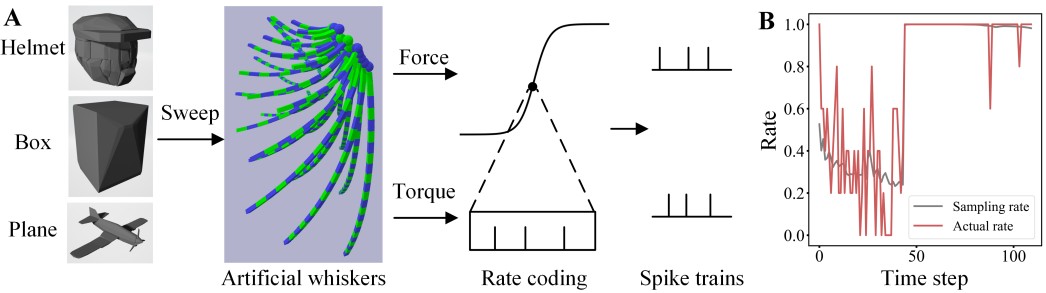

Figure 3: **Spiking whisker sweep dataset.** (**A**) 3D objects sweep across the virtual whisker array in a physics engine (Zhuang et al., 2017), recording the force and torque signals from whisker base sensors. Then, the analog inputs are converted into spiking inputs by generating spike trains within a time window through the sigmoid function and Bernoulli sampling. (**B**) Bernoulli sampling probability calculated by the sigmoid function and the actual generated spikes firing rate.

tegration and processing in the brain. Advances in the understanding of cell types, locations, and projection patterns (Peron et al., 2015; Sermet et al., 2019; Zhang & Zagha, 2023; Estebanez et al., 2016; Gardères et al., 2024) in the mouse barrel cortex indicate that the sensor-motor integration in biological systems possesses detailed anatomical principles.

**Biologically constrained model for barrel cortex:** Drawing upon the continuously accumulating quantitative anatomical data, neuroscientists have attempted to reconstruct local circuitry of the barrel cortex on silicon, aiming to validate our understanding of the sensory system. For instance, by employing the Izhikevich single-neuron model (Izhikevich, 2003), a three-barrel model of the superficial layer was constructed to mimic the alterations in synaptic strength that occur subsequent to whisker deprivation (Huang et al., 2022). The dendritic model of spiny stellate cells (Lavzin et al., 2012), direction selectivity model (Kremer et al., 2011), and layer 4 local network model in Fmr1-KO mice (Domanski et al., 2019) have also been constructed. These biological computing models accurately reflect the complex activities in the sensory system, yet they lack perceptual functions in behavioral contexts. This limitation stems primarily from two key factors: Firstly, the intricate dynamics and inherent instabilities of biological models, such as neuronal refractory periods, are not conducive to computational gradients. Secondly, the substantial cost associated with live experiments and the challenges in precisely capturing sensory signals have led to a scarcity of datasets, which in turn, hinders the direct training of biological models akin to machine learning paradigms. Consequently, the application potential of these biological models remains constrained.

**Trainable biologically constrained model:** However, mainstream trainable models overlook the detailed structure of cortical circuits. For instance, artificial neural networks typically adopt regular fully-connected (multilayer perceptrons) or locally-connected (convolutional neural networks) architectures, which are discrepant from the intricately heterogeneous connectivity found in real cortex (Oh et al., 2014). Furthermore, current models do not strictly differentiate between neuron types (excitatory or inhibitory) and numbers, which have been a focal point in recent neuroscientific efforts to decipher brain function (Muñoz-Castañeda et al., 2021; Yao et al., 2023). Therefore, an emerging direction is to integrate more detailed biological mechanisms into traditional computational models, such as introducing dendrites to enhance the spatiotemporal dynamics of network models (Zheng et al., 2024). A spiking neural network (Chen et al., 2022), consisting of over 50,000 neurons distributed across four distinct subtypes, was trained on five biologically pertinent visual tasks, leveraging the findings from neuroscience on the mouse primary visual cortex (Billeh et al., 2020). However, to our best knowledge, network models that possess both the topological structure of the sensory cortex and perceptual behavioral functions are currently rare.

## 3 METHODOLOGY

### 3.1 MODEL CONSTRUCTION

**Network topology:** To enhance biological authenticity, we established a biologically constrained barrel cortex model with reference to the study in neuroscience. The number, location, type, and con-

nectivity of neurons in our model faithfully replicate the findings from the anatomical study (Huang et al., 2022). This study utilized immunohistochemistry, confocal imaging, and other biological techniques to elucidate specific neuronal distribution patterns in the superficial layers of the mouse barrel cortex. In general, our model comprises 13 neuronal subtypes, including 3 excitatory and 10 inhibitory subtypes, totaling 4,218 neurons, along with an additional 200 thalamic neurons. Following the classical "whisker-barrel" signaling pathway (see in Fig. 1A, C), external inputs are initially processed by 200 thalamic neurons and then primarily propagate to four neuronal subtypes found in layer 4 (L4) of the barrel cortex. Considering that the projecting neurons are predominantly excitatory in brain, we use the state of three excitatory communities processed through two linear layers as the readout for the entire network (see in Fig. 1D).

The number of connections between neurons is generated through sampling based on experimentally measured projection strength (see in Fig. 2A). Assuming there are $n$ presynaptic neurons and $m$ postsynaptic neurons, the potential number of connections is $n \times m$. We introduce a boolean matrix $M$ to constrain the number of connections. The calculation formula is as follows:

$$W^{'} = W \odot M, W \in \mathbb{R}^{n \times m}$$

$$\frac{1}{nm} \sum_{i=1}^{n} \sum_{j=1}^{m} M_{ij} = p, M_{ij} \in \{0, 1\}, p \in [0, 1] \tag{1}$$

$$PMF(M_{ij}) = p \cdot \delta_{M_{ij},1} + (1 - p) \cdot \delta_{M_{ij},0}$$

, where $W$ and $W^{'}$ denote the original and constrained connection weight matrices, respectively. $M$ represents a boolean matrix of the same size as $W$, where an element of 1 indicates the presence of a corresponding connection, and 0 indicates its absence. $p$ and $\delta$ stand for the experimentally measured connection probability and Kronecker delta, respectively. $PMF$ signifies the Probability Mass Function. Weights and gradients are only computed and updated on existing connections. By repeating this process to establish all connections, our model can be collectively viewed as a spiking recurrent neural network, with each neuronal population construed as a hidden layer within it.

**Single-neuron dynamic model:** We employ the adaptive Leaky Integrate-and-Fire (aLIF) (Yin et al., 2020) model as the backbone for depicting single-neuron dynamics. In comparison to the frequently employed Izhikevich (Izhikevich, 2003) and GLIF (Teeter et al., 2018) neural models in neuroscience, the aLIF model offers simplified computational requirements and is more amenable to training. In contrast to the straightforward LIF neurons, the aLIF model sets the time constant as trainable parameters, thereby effectively capturing the dynamic variations among neurons within cortical networks. The membrane potential $V$ at time $t$ is updated according to this formula:

$$V(t) = e^{-\frac{1}{\tau_m}} \cdot V(t-1) + (1 - e^{-\frac{1}{\tau_m}}) \cdot R \cdot I(t) - S(t-1) \cdot \theta(t-1) \tag{2}$$

, where $R$ and $\tau_m$ respectively denote the resistance and time constant of cell membranes, with $I$ representing the input current. $S$ is a heaviside function used to indicate whether the neuron emits a spike. If a spike is emitted, the membrane potential $V$ undergoes a soft reset by subtracting the current firing threshold $\theta$.

To avoid abnormal firing, the fired spike triggers a temporary increase in the firing threshold $\theta$. The firing threshold $\theta(t)$ at time $t$ can be calculated using the following formula:

$$\theta(t) = \theta_{init} + \beta \cdot \eta(t)$$

$$\eta(t) = e^{-\frac{1}{\tau_{adp}}} \cdot \eta(t-1) + (1 - e^{-\frac{1}{\tau_{adp}}}) \cdot S(t-1) \tag{3}$$

, where $\theta_{init}$ is the minimum firing threshold. $\beta$ is a constant that regulates the increment of the threshold induced by spikes $\eta$, which we set to 1.8. $\tau_{adp}$ represents the adaptive time constant of the firing threshold. Notably, we set $\tau_m$ and $\tau_{adp}$ as two additional trainable parameters to investigate the differentiation of neural dynamics in the trained model (see in Sec. 4.3).

To reflect neuronal diversity, we have additionally specified the categories of input current to the neurons: excitatory or inhibitory currents. Neurons in the network receive not only external input currents $I_e$ but also currents $I_{rec}$ from connected neurons. The total current $I(t)$ received by a neuron at time $t$ can be calculated by the following formula:

$$I(t) = W_e I_e(t) + I_{rec}(t), W_e > 0$$

$$I_{rec}(t) = \sum_{i} H_i W_i^{rec} S_i(t-1), H_i \in \{-1, 1\}, W^{rec} > 0 \tag{4}$$

Table 1: **Classification accuracies (%) among models.** ANNs are tested on the original real-valued whisker sweep dataset (Zhuang et al., 2017), while SNNs are tested on both the real-valued dataset and the spiking-based dataset constructed in Sec. 3.3.

| ANNs | ST CNN (Zhuang et al., 2017) | DB LSTM (Ullah et al., 2017) | RNN+ (Collins et al., 2017) | UGRNN |
|---|---|---|---|---|
| Real-valued | 67.2 | 70.9 | 78.2 | 76.4 |
| SNNs | SRNN 256 (Yin et al., 2020) | STSC SNN (Yu et al., 2022) | TA SNN (Yao et al., 2021) | Barrel model (ours) |
| Spiking-based | **81.8** | 80.0 | **81.8** | **81.8** |
| Real-valued | 85.5 | 87.3 | 83.6 | **89.1** |

Table 2: **Classification accuracies of SNNs in whisker deprivation experiments.** Tested on the spiking whisker sweep dataset.

| SNNs | Number of deprived whiskers | | | | | | | | |
|---|---|---|---|---|---|---|---|---|---|
| | 0 | 1 | 2 | 3 | 4 | 5 | 6 | 7 | 8 |
| SRNN 256 (Yin et al., 2020) | **81.8** | 69.0 | 66.2 | 61.6 | 59.0 | 57.0 | 55.0 | 53.4 | 53.2 |
| STSC SNN (Yu et al., 2022) | 80.0 | 71.9 | 68.4 | 64.8 | 65.2 | 61.7 | 58.3 | 57.5 | 57.2 |
| TA SNN (Yao et al., 2021) | **81.8** | 72.4 | 66.3 | 64.5 | 60.4 | 56.8 | 54.7 | 47.8 | 48.4 |
| Barrel model (ours) | **81.8** | **73.8** | **69.7** | **67.9** | **66.1** | **63.7** | **60.4** | **60.1** | **58.0** |

, where $W_e$ and $W_i^{rec}$ represent the weights of the external connection and the connection with the $i$-th presynaptic neuron, respectively. $H_i$ is a binary constant used to indicate the type of the $i$-th presynaptic neuron, where 1 represents an excitatory type and $-1$ represents an inhibitory type. $S$ represents the spike emitted by the $i$-th presynaptic neuron. To distinguish the effects of excitatory and inhibitory currents, all connection weights are clamped to remain positive.

Based on the aforementioned single-neuron dynamics, we can apply Backpropagation-Through-Time (BPTT) to train the entire network. To compute the gradients at the spike-timing, we employ a Gaussian function as the surrogate gradient:

$$grad = \mathcal{N}(V(t)|\theta(t), \sigma^2) \tag{5}$$

, where the mean of distribution $\mathcal{N}$ is the firing threshold $\theta$, and the variance is a constant $\sigma$ used to scale the membrane potential. Unless otherwise specified, the parameters of neural dynamics in our network are set to the same value.

## 3.2 MODEL TRAINING

Another primary issue in training biologically constrained models is that, as a complex dynamical system, subtle variations in initial parameters can lead to bifurcation in our biological network (Wang, 2022). Therefore, searching for appropriate initialization parameters is crucial for stable training of the network state. Here, we investigate two critical parameters: the initial firing threshold $\theta_{init}$ and the initialized connection weight $W_{init}$. After generating neuronal connections according to Eq. 1, all connection weights are randomly sampled from a uniform distribution within the range $[0, W_{init}]$.

Chaotic neuronal activity, as opposed to regular firing, typically signifies a more abundant learnable representational space. Consequently, we introduce the coefficient of variation (CV) as an order parameter to quantify the synchronization of neuronal electrical activity (Tian et al., 2022). Prior to training, we first input samples from the dataset into networks with varying initial states, and quantify the activity of all neurons during the forward propagation process. For an initial network, the spike emissions during the forward propagation of external inputs with time length $T$ among $N$ neurons are recorded, forming an $N \times T$ boolean matrix. Then coefficients of variation can be calculated with this formula:

$$CV = \frac{1}{N} \sum_{i=1}^{N} \frac{\sigma_i}{\mu_i} \tag{6}$$

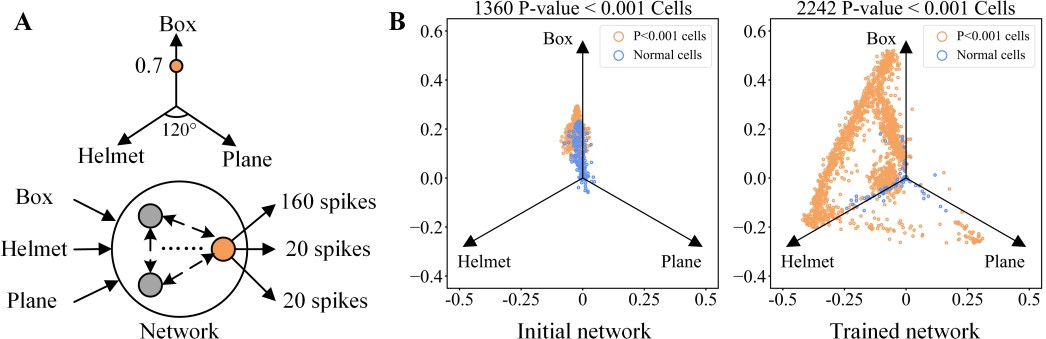

Figure 4: **Firing selectivity of neurons in the trained network.** (**A**) Count the spikes for each neuron across three categories of inputs, and use this as a vote to determine its displacement in the directions representing the three categories. (**B**) Visualization results of input-specific firing in the initialized network (left) and the trained network (right). Neurons with a P-value less than 0.001, as determined by one-way ANOVA, are considered to exhibit significant firing selectivity.

, where $\mu_i$ and $\sigma_i$ represent the mean firing rate and standard deviation, respectively, of the $i$-th neuron. We consider the average value of CV across all input samples as the final score for this initial network. Fig 2B illustrates the evolution of CV score, where a higher value typically indicates more disordered neuronal activity (see in the supplementary material). Fig 2C presents the training performance of three differently initialized networks on the dataset constructed in Sec 3.3, with the initialization exhibiting the higher CV score yielding the best training performance.

## 3.3 SPIKING WHISKER SWEEP DATASET

The absence of corresponding behavioral datasets poses a significant hindrance to the training of biologically constrained models. High time cost of in vivo behavioral experiments and difficulty in accurately recording whisker signals have resulted in a scarcity of available datasets that directly correspond to the barrel cortex. As a result, we converted a virtual whisker sweep dataset (Zhuang et al., 2017) into the spiking format, which serves as a platform for training and analyzing our barrel model, while also facilitating comparisons with mainstream computational models.

In the field of sensory-motor integration in neuroscience, scientists have leveraged anatomical knowledge of rat heads and whiskers to create a physically and biologically realistic whisker array model within a physics engine (Coumans & Bai, 2016–2021). This model comprises 31 whiskers, accurately reflecting mechanical properties such as length, intrinsic curvature, and relative positions. By recording the forces and torques experienced by each whisker as three-dimensional models from ShapeNet dataset (Chang et al., 2015) sweep across the whisker array, a virtual whisker signal classification task can be established.

However, this whisker sweep dataset was originally created for training convolutional neural networks, and thus requires converting signals into spiking-based inputs before they can be used for our biologically constrained network training. Recent neuroscientific research has revealed that whisker vibration signals undergo a transformation from temporal coding to cortical rate coding in the thalamus (Lee et al., 2024). Inspired by this finding, we have developed a method to generate spike trains from force and torque signals recorded by sensors, utilizing a rate coding approach (see in Fig. 3). For the magnitude of the force detected by a whisker sensor at a particular moment $|F|$, we map it to a probability $p$ of firing a pulse using the sigmoid function:

$$p = \frac{1}{1 + e^{-(\ln |F| - c)}} \tag{7}$$

, where $c$ is the shift coefficient. Subsequently, based on the probability $p$, independent spikes are generated within a fixed-length time window using Bernoulli sampling. We set the length of time windows to 5, meaning that the time step is extended to five times its original length. The sampling probability versus the actual pulse firing rate is illustrated in Fig. 3B. The torque is processed in the same manner as the force. Specifically, the shape of a single sample in our spiking whisker sweep dataset is $(550, 31, 3, 2)$, representing 550 time steps, 31 whiskers, 3 sensors at the base of each

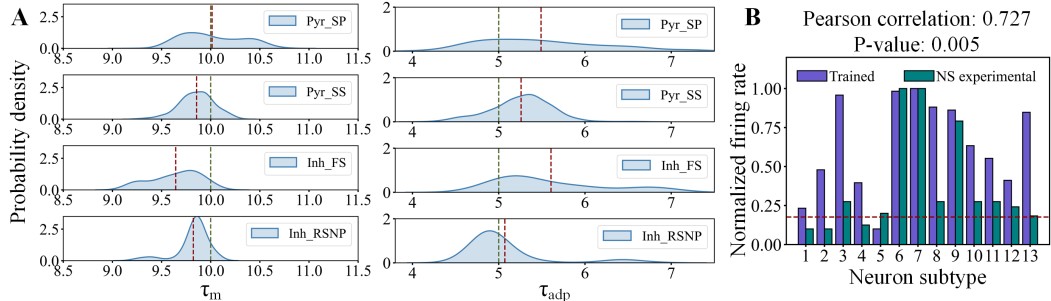

Figure 5: **Emergent similarities in neural dynamics between the trained network and neuro-science experiments.** (**A**) Kernel density estimation plots of $\tau_m$ and $\tau_{adp}$ for the four L4 subtypes in the trained network. Red vertical lines denote the mean value, while green vertical lines indicate the identical initial values assigned. The complete set of 13 subtypes is presented in the supplementary material. (**B**) The comparison of the 13 subtypes after training with experimental-based firing rates, quantified as the Pearson correlation coefficient. The firing rates are normalized to the interval $[0.1, 1]$ to draw the chart. Due to the identical initial dynamic parameters of all neurons, the firing rates are the same pre-training (red reference line).

whisker, and 2 channels for force and torque, respectively. There are 91 samples for each of the three classes, and the dataset is randomly split into training and testing sets in an 8:2 ratio.

## 4 EXPERIMENTAL RESULTS

In this section, we compared comprehensively the ability of our model to perceive objects with several mainstream computational models based on the whisker sweep dataset. Then, we emphasize the biological interpretability of our trained model, delving into the dynamic evolution that transpires within our model throughout the training phase. This exploration offers insights into how the model's behavior mimics and elucidates underlying biological mechanisms. Details of the model training are provided in the Appendix.

### 4.1 PERFORMANCE COMPARISON WITH BASELINE MODELS

We first evaluated the ability of our barrel cortex constrained model to distinguish objects based on whisker signals and contrasted its classification accuracy with other prominent computational models. As shown in Tab. 1, our model's accuracy on the whisker sweep dataset surpassed traditional ANNs, reaching a peak of 89.1% on the original real-valued dataset and matching the best performance of recent SNNs at 81.8% on our spiking-based dataset.

Subsequently, we introduced a commonly used whisker deprivation experiment from neuroscience to further compare the whisker-related perceptual abilities of our model with other SNNs. Neuroscientists commonly trim or cauterize specific whiskers of mice in behavioral experiments to study the plasticity of sensory neural system (Gunner et al., 2019; Pan et al., 2022). We mimicked this scenario by randomly occluding different numbers of whisker signals in the spiking whisker sweep dataset (experimental details are provided in the Appendix). The results (see in Tab. 2) demonstrate that our model consistently achieved the highest classification accuracy in all experimental settings, showcasing not only improved fidelity in sensory circuitry but also perceptual classification capabilities that rival those of mainstream computational models.

### 4.2 NEURAL SPECIFIC FIRING SELECTIVITY

Next, we focus on examining the internal modifications that transpire within our model as it develops perceptual capabilities, starting with the firing selectivity. The selective firing observed by neuroscientists in the sensory cortex is a prominent feature for elucidating the mechanisms of sensory functions in the brain (Guy et al., 2023). In vivo experiments have revealed that the barrel cortex contains a vast range of neurons with specific responses; these neurons exhibit strong reactions to

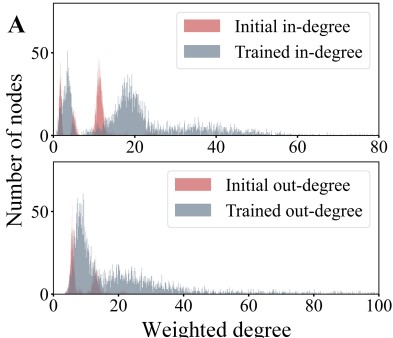
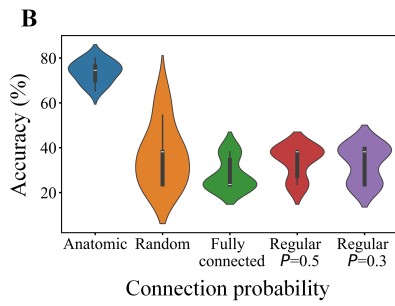

Figure 6: (**A**) Alterations in weighted degree distribution pre- and post-training. (**B**) Training performance of networks created with varying connection probabilities. From left to right: anatomically based connection probabilities, random connection probabilities, fully connected network, and fixed connection probability $P$. Each configuration was trained ten times to plot the violin diagram. All connection configurations, with the exception of random connections, were initialized using the method described in Sec. 3.2, with the process detailed in the supplementary material.

signals of particular directions (Wilent & Contreras, 2005; Estebanez et al., 2016) and deflection angles (Lavzin et al., 2012) but are relatively insensitive to irrelevant signals.

The approach for quantifying neuronal firing selectivity in our model illustrated in Fig. 4A. Specifically, we sequentially input the three types of samples from dataset into the network, count category-related spikes for each cell, and then employ one-way Analysis of Variance (ANOVA) to judge whether significant differences exist. A neuron is considered input-specific if the resulting P-value is less than 0.001. To visualize this category selectivity, we divided three equally spaced directions and voted for its position according to the number of spikes by cells respond to each category.

Statistical results show that after training, the proportion of neurons with significant firing selectivity has increased from 32.2% to 53.2%. Moreover, the differentiation of category preference becomes more evident (see in Fig. 4B). In the initialized network, the number of spikes responded by neurons to different categories of inputs is balanced, so they are clustered in the central area. However, in the trained network, a cluster of neurons is gathered in the direction of each category, indicating that the number of spikes released in response to that category of input far exceeds that of the other two categories. The same selective firing analysis for other SNNs can be found in the supplementary material. Reproducing neural firing selectivity similar to that of brains enhances the biological interpretability of our network.

### 4.3 DIFFERENTIATED NEURONAL DYNAMICS

Furthermore, we compared the dynamical similarity of 13 neural subtypes between our trained network and those statistically observed in neuroscience experiments, thereby quantifying the biological plausibility of our model. The heterogeneous dynamical properties across neocortex serve as the structural foundation for the coordinated function of different regions (Wang, 2022). Although all neuronal subtypes were initially assigned the same values for $\tau_m$ and $\tau_{adp}$ elaborated in Sec. 3.1, during the training process, these two trainable parameters self-updated in response to optimizing the objective function, leading to the differentiation of individual dynamics for each neuron.

Fig. 5 demonstrates that our network, under the optimization constraint of the classification task only, spontaneously differentiates into dynamic gradients across 13 neuronal subtypes (see in Fig. 5A). Then, we quantitatively compared the similarity between our neural dynamic gradients and experimental results (see in Fig. 5B). Corresponding to these neuron subtypes, Izhikevich neuronal models (Izhikevich, 2003) have been established in neuroscience to fit the data from biological experiments (Huang et al., 2022). We preserved the dynamic parameters of all neuronal subtypes post-training and then applied an identical continuous 550 ms, 1mV constant current to our neurons and their corresponding Izhikevich neurons, recording their firing rates and comparing their similarity. The external current was applied in isolation to each subtype, rather than across the entire network, to prevent cross-contamination of effects. Then, we calculate the Pearson correlation co-

efficient between these two firing rates. In the initial network, all neurons were assigned identical dynamic parameters, resulting in equal firing rates (red reference line in Fig. 5B) across subtypes and no Pearson correlation. However, after training, the Pearson correlation coefficient increased to 0.73 with a P-value of 0.005. Given that biological neuron models typically have longer simulation steps compared to aLIF neurons, we employed a simple normalization to scale their firing rates to the same range $[0.1, 1.0]$ to better visualize them together. The normalization process is as follows:

$$fr^{'} = 0.1 + 0.9 \times \frac{(fr - fr_{min})}{(fr_{max} - fr_{min})} \tag{8}$$

, where $fr$ and $fr^{'}$ represent the original and normalized firing rates. $fr_{max}$ and $fr_{min}$ denote the highest and lowest firing rates among 13 neural subtypes. This normalization does not affect the calculation of the Pearson correlation coefficient. The results indicate that our trained network spontaneously developed dynamic gradients analogous to those observed in the real barrel cortex.

### 4.4 Degree Distribution and Connectivity Probability Ablation

Finally, we investigated the alterations in the network's topological structure through statistical analysis of the weighted degree distribution among its nodes. Contrary to the abstract neurons in machine learning models, each neuron in our model is a well-defined biological entity, allowing our network to be viewed from a graph-theoretic perspective (Lynn & Bassett, 2018). We define the sum of the weights of neuron-related connections as the weighted in-degree and out-degree, and then tally the number of nodes for each weighted degree.

Results show that both the in-degree and out-degree distribution of our network exhibit a pronounced long-tail effect after training (see in Fig. 6A), indicating that the network has spontaneously evolved important hub nodes during training process. The degree distribution of neurons in the network is more dispersed than initially, which is consistent with reports on the scale-free properties of brain networks in related work (Zheng et al., 2020; Henriksen et al., 2016; Yu et al., 2018). The two general conditions for the generation of scale-free networks are model growth and preferential attachment (Henriksen et al., 2016). However, our network can also reproduce similar properties merely by optimizing the weights of existing connections through training, which provide a new direction for establishing structurally realistic brain network models in neuroscience. The results using the same statistical method on other models can be found in the supplementary material.

We additionally quantified the impact of anatomic connection probabilities within our network, comparing its training performance to networks established with random connections, fully connected network and regular connection probabilities of 0.5 and 0.3. The results suggest that networks which maintain the anatomic connection probabilities are not only the most biologically authentic but also the most effective to train in our whisker-related sensory task (see in Fig. 6B).

## 5 Conclusion

In this paper, we develop an effective constructing and training pipeline for biologically constrained sensory-motor integration networks. Our model, grounded in the authentic anatomical structure of the mouse barrel cortex, matches mainstream computing models in sensory performance and spontaneously aligns with the firing characteristics, dynamics, and degree distribution observed in real neural systems during the training process. Through the innovative pipeline and converted spiking-based whisker sensory dataset, we advanced the bridging of the gap between biological realism and computational performance in current sensory related models.

**Limitations and future work:** In the initialization process, we introduced the Coefficient of Variation (CV) measure as a general principle, but a more granular analysis depends on further integration with research outcomes from brain dynamics. Figure 1 in the supplementary material presents raster plots of our model, where abnormal synchronous firing (with a higher CV score indicating stronger intensity) can still be observed in the initial network. This is due to all neurons being assigned same parameter values at the start. Although this does not affect the training and disappears in the trained model, it reduces the biological authenticity of the initial network. Further constraining neuronal activity can bring it closer to biologically realistic networks. For example, there is research potential in employing more nuanced biological constraints, such as membrane potential dynamics (Rossbroich et al., 2022) and synaptic scaling, to guide the initialization.

ACKNOWLEDGMENTS

We thank Dr. Chengxu Zhuang for helping us reproduce the whisker sweep dataset. This work was financially supported by the STI 2030-Major Projects (2021ZD0201002), and National Natural Science Foundation of China grants (T2122015, 32471149).

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

## A   APPENDIX

### A.1   IMPLEMENTATION DETAILS

We employ Backpropagation-Through-Time (BPTT) to train our network, adopting the cross-entropy loss function. Our network model was trained for 250 epochs using the Adam optimizer, with an initial learning rate of 0.08, which was decayed by a factor of 0.8 every 20 epochs. The batch size was set to 128. $\theta_{init}$ and $W_{init}$ discussed in section 3.2 were set to 0.04 and 0.06 respectively. All experiments were carried out on a single NVIDIA A100 80 GB GPU. For the artificial neural networks (ANNs) under comparison (Zhuang et al., 2017; Collins et al., 2017; Ullah et al., 2017), we adhere to the existing experimental setup (Zhuang et al., 2017), utilizing the original real-valued data inputs. In the experiments presented in Tab. 1, spiking neural networks (SNNs) (Yu et al., 2022; Yin et al., 2020; Yao et al., 2021) are tested on both the original real-valued dataset and the spiking-based dataset devised in Sec. 3.3. Additionally, unless otherwise specified, other experiments of our model are based on the spiking-based dataset.

**Whisker deprivation:** We randomly set varying numbers of whisker signals in the test set samples to zero and recorded the accuracy of trained models. This process was repeated multiple times, and the average accuracy was presented as the results in Tab. 2. The experiment of depriving one whisker was repeated 31 times, while the other experiments were each repeated 50 times.

