# OpenReview forum: "Biologically Constrained Barrel Cortex Model Integrates Whisker Inputs and Replicates Key Brain Network Dynamics"
_ICLR.cc/2025/Conference — ICLR 2025 Spotlight_

### Official Review · Reviewer_tE3s · 2024-10-19

**Soundness:** 3
**Presentation:** 4
**Contribution:** 3
**Rating:** 8
**Confidence:** 3

**Summary:**

The paper presents a novel biologically constrained spiking network model of the mouse barrel cortex, introduces an initialization technique for it based on the coefficient of variation of the network activity, and evaluates both in conjunction on a newly spike converted whisker sweep dataset. The proposed method shows favorable performance compared to other even non-constrained network models, and fully trained versions of it more closely reproduce biologically observed network characteristics such as increased neuron selectivity, neuron type dependent firing rate and timescale distribution, and neuron weighted degree distribution.

**Strengths:**

- The paper is really well written with predominently precise use of terminology.
- It tackles an interesting and really challenging problem; effective training of biologically constrained networks, and computationally characterizing them.
- Achieving competitive performance on benchmarks given this many biological constraints, even if potentially computationally more expensive, is no easy feat.
- Also impressive is the sheer number of directions in which training the networks increased the similarity to biological network statistics.
- The dimensions along which the network was characterized are well motivated, and the used quantitative measures are really creative and nicely connected back to neuroscience literature.

**Weaknesses:**

That the training methodology is really more efficient was not convincingly shown, and to call it a training framework might be a stretch:
   * 1.1) An effective initialization technique was shown (the coefficient of variation approach), yet no qualitative assessment of efficiency, such as Performance/FLOPs or Performance/Samples is provided to warrant the statement of “efficient” from my understanding.
   * 1.2) Also consider, that figure 2C doesnt even show the efficacy convincingly, as initially all three visualized initializations perform similarly well (until around epoch 20), and subsequently run into training stability issues (examplified by the sudden drop/rise in performance). In fact even the supposed best initialization (according to the coefficient of variation) displays two mayor drops in performance during training. This could be more indicative of exploding gradient issues or extreme parameter regimes, which would rather be fixed by gradient clipping or appropriate regularization and normalization techniques: no mention of such things in the methods?
   * 1.3) Furthermore in panel 2D the network displays pathological firing synchronization in both cases, unless I’m missing something (how does a biological recording from that brain region typically look like?). This could be indicative of a lack of decorrelating mechanisms or insufficient inhibition, which to my knowledge the brain typically has.
   * 1.4) Lastly, no comparison to other initialization techniques typical in the spiking literature was conducted, such as simply scaling synapse weights by in-degree, or more sophisticated ones (e.g. see Rossbroich et. al 2022).
* However, somewhat reformulating the claims and contributions probably is sufficient to make the paper consistent.

The computational characterizations consistently lacks a control / comparison to interpret the results, and are possibly merely side effects of simple gradient descent training in networks?
   * 2.1) For neuron specific firing selectivity: it would be interesting to see how the CNNs or LSTMs neuron specificity (or similar measure) behaves? If this doesnt happen for them it’s more convincing that this is because of the biological constraint and not simply due to generic gradient optimization.
   * 2.2) For the differentiated neuronal dynamics: when initializing to a single value and training with backprop naturally the distribution become smudged, and when margenalizing over neuron type there will naturally be differences. Whether these differences here are statistically significant from random drift, or whether they reflect more closely the biological ground truth, remains unclear from the paper? The firing rate example is more convincing, yet the effect might be primarily driven by only three subtypes (6,7 and 9) and the impact of normalizing the firing rates is unclear to me. Some sort of additional control would further substantiate the point.
   * 2.3) For the weight degree distribution: one-sided metrics evaluated on variables with low variance will typically also display a longer tailed distribution in networks after training, as the variable gets smudged and the interval is only open to one side. Without a control of sorts it’s hard to understand whether in this case this is an effect of the cortical constrains or simply due to backpropagation and this being a directed graph. Evaluating this or a similar metric for e.g. the LSTM could possibly serve as a control / comparison here too.
   * 2.4) Regarding the connectivity probability: this is more convincing, yet only IFF the network was properly re-initialized for the changed connectivity, e.g. using the proposed coefficient of variation technique.
   * However, it probably cannot be expected to also get explanations as to “why” we see these effects in a single paper, and therefore an extension of the future work section and making ther readers aware of these possible limitation should be enough to address most of these issues.

**Questions:**

1) The firing activity in figure 2D seem very synchronized and periodic, does this reflect real work data, and do you have example visualization of such data?
2) Any explanation as to why the anatomic connection probability is best? Were the initialization adjusted individually for each of the connection probabilities?
3) For each of the network characteristics findings; do you believe you find this effect due to the biological constrains or simply gradient based training in networks?
4) Where any normalization or regularization techniques used for training, or gradient clipping?

---

> ### Author Response · Authors · 2024-11-23
> **Response to Reviewer tE3s (part 1)**
>
> Thank you for taking the time to review. Please see our detailed response to the issues you have raised.
> ## Response to Weaknesses
> > W1.1: An effective initialization technique was shown (the coefficient of variation approach), yet no qualitative assessment of efficiency, such as Performance/FLOPs or Performance/Samples is provided to warrant the statement of “efficient” from my understanding.
>
> **A1:**  Thank you for your concern regarding the phrasing in our work. We would like to clarify that the term "efficient" is used in the context of training biological neural networks, including the construction of biologically constrained models and training strategies. We use the term "efficient" for three main reasons:
> 1. As you mentioned, training biological neural networks is a widely acknowledged challenge, currently at a stage where success is binary (Yes or No), far from the optimization of computational costs seen in ANNs. Our work is dedicated to addressing this pre-existing challenge.
> 2. To our best knowledge, the only prior work, Chen et al., 2022, as declared in their paper, trained a V1 network with each training session required 60 hours on 160 A100 GPUs, a cost that is beyond the means of many laboratories. In contrast, our work relies on a single A100 GPU, significantly reducing the hardware requirements for research.
> 3. Our method employs minimalistic techniques to construct and train the barrel cortex model, including the aLIF model, connection probabilities, and the CV measure, without incorporating any additional complex tricks. **We believe that the more straightforward and intuitive the method, the stronger its generalizability**, and thus its ease of extension to other neural circuits.
>
> This is why we describe our approach as efficient, and we look forward to future research that can continue to reduce the costs associated with training biological neural networks.
>
> > W1.2: Also consider, that figure 2C doesnt even show the efficacy convincingly, as initially all three visualized initializations perform similarly well (until around epoch 20), and subsequently run into training stability issues (examplified by the sudden drop/rise in performance). In fact even the supposed best initialization (according to the coefficient of variation) displays two mayor drops in performance during training. This could be more indicative of exploding gradient issues or extreme parameter regimes, which would rather be fixed by gradient clipping or appropriate regularization and normalization techniques: no mention of such things in the methods?
>
> **A2:** Thank you for raising the issue of training stability; we have addressed it by reducing the learning rate. We have adjusted the learning rate decay to halve every 10 epochs, which has smoothed the training process (**see in revised Figure 2**). As we employed an early stop, the previous fluctuations in the training process did not affect our model's performance.
>
> > W1.3: Furthermore in panel 2D the network displays pathological firing synchronization in both cases, unless I’m missing something (how does a biological recording from that brain region typically look like?). This could be indicative of a lack of decorrelating mechanisms or insufficient inhibition, which to my knowledge the brain typically has.
>
> **A3:** Thank you for raising the issue of abnormal synchronous firing; we have additionally presented raster plots in the trained network to illustrate more biologically plausible neural activity. The pathological synchronous firing you mentioned originates from the initialized network, where all neurons are assigned identical initial parameters. Additionally, the aLIF model is more prone to firing compared to biological neurons, such as the Hodgkin-Huxley model, which reflects a common distinction between spiking neural networks and biological neural networks.
>
> To provide a more comprehensive response to your issue, we have provided raster plots of the trained network (**see Figure 1 in the supplementary material**), which demonstrate the disappearance of pathological synchronous firing. Concurrently, we have presented the firing rates of various neuron types, with inhibitory neurons exhibiting higher firing rates than excitatory neurons, aligning with previous neuroscientific research on the excitatory-inhibitory balance in barrel cortex (e.g., "Synaptic Computation and Sensory Processing in Neocortical Layer 2/3," 2013, and "A Cellular Resolution Map of Barrel Cortex Activity during Tactile Behavior," 2015).

---

> ### Author Response · Authors · 2024-11-23
> **Response to Reviewer tE3s (part 2)**
>
> This is a continued response to the weaknesses 1.4-2.1.
> ## Response to Weaknesses
> > W1.4: 1.4) Lastly, no comparison to other initialization techniques typical in the spiking literature was conducted, such as simply scaling synapse weights by in-degree, or more sophisticated ones (e.g. see Rossbroich et. al 2022).
>
> **A4:** Thank you for considering a comparison with other initialization techniques. **There are three direct reasons why our network cannot directly employ conventional initialization methods** such as Kaiming or Xavier initialization: **1.** To distinguish between excitatory or inhibitory pre-synaptic currents, we have constrained all weights to be always positive (see in Equation 4). **2.** We aim to concentrate all synaptic weights within a similar small range to highlight the impact of anatomical connections. **3.** We wish to control the connection strength across the entire network through a single parameter to study the network's various states. Therefore, we sample all initial synaptic weights from a uniform distribution [0, W_init].
>
> However, the article you mentioned, Rossbroich et al., 2022, holds promise for integration with our work. They initialize parameters by simulating biologically plausible membrane potential dynamics, which can be effectively integrated with neuroscience research (for example, "Membrane potential dynamics of excitatory and inhibitory neurons in mouse barrel cortex during active whisker sensing," 2023), and is expected to address the issue of membrane potential initialization in our study (currently, we utilize random initial membrane potentials). But, this is almost another research topic in itself. Consequently, **we discussed and cited this work in the conclusion section.**
>
> > W2.1: For neuron specific firing selectivity: it would be interesting to see how the CNNs or LSTMs neuron specificity (or similar measure) behaves? If this doesnt happen for them it’s more convincing that this is because of the biological constraint and not simply due to generic gradient optimization.
>
> **A5:** Thank you for the crucial suggestion to compare firing selectivity with other models. **We have conducted analogous experiments on three other SNNs, and the results indicate that this is a distinctive feature of our model.**
>
> Given that CNNs and LSTMs are not well-suited for the same method of spike statistics as described in Section 4.2, we have supplemented our study with experiments on the selective firing of other three SNNs mentioned in the manuscript (**see Figure 3 in the supplementary material**). It can be observed that both the SRNN 256 and TA SNN do not exhibit significant firing selectivity before or after training, indicating that the firing selectivity is not a universal phenomenon.
>
> There are at least three significant differences compared to our model, despite numerous neurons in the STSC SNN demonstrated preferred firing: **1.** Our model did not have significant firing selectivity in its initial state; the preferential firing emerged spontaneously during the learning of a sensory function. In contrast, approximately half of the neurons in the STSC SNN showed selectivity from the outset. **2.** The firing preferences in our model are more concentrated, forming distinct functional clusters, whereas the STSC SNN shows a more uniform distribution. **3.** Fundamentally, each neuron in our model can be correlated with a biological neuron in the cortex, offering the potential for further comparison with actual neural circuits, a feature that conventional SNNs lack in terms of biological interpretability.

---

> ### Author Response · Authors · 2024-11-23
> **Response to Reviewer tE3s (part 3)**
>
> This is a continued response to the weaknesses 2.2-2.3.
> ## Response to Weaknesses
> > W2.2: For the differentiated neuronal dynamics: when initializing to a single value and training with backprop naturally the distribution become smudged, and when margenalizing over neuron type there will naturally be differences. Whether these differences here are statistically significant from random drift, or whether they reflect more closely the biological ground truth, remains unclear from the paper? The firing rate example is more convincing, yet the effect might be primarily driven by only three subtypes (6,7 and 9) and the impact of normalizing the firing rates is unclear to me. Some sort of additional control would further substantiate the point.
>
> **A6:** Thank you for your concern regarding the biological plausibility of the differentiated neuronal dynamics. **We have validated the similarity to the biological ground truth by comparing the firing rates with those of real neuronal subtypes as depicted in Figure 5B**, and the results show a Pearson correlation coefficient of 0.66 with data based on neuroscientific experiments. Below is the detailed elucidation of our methodology.
>
> Initially, each neuron in our model was assigned identical initial parameters, therefore a consistent response to identical input currents (red reference line in Figure 5B). After training, we preserved the differentiated parameters for each neuronal subtype within our network. Subsequently, a sustained 1mV external current was applied to each neuron within the subtypes for a duration of 550ms, and the total spikes count for the entire subtype was tallied. This was conducted in isolation for each subtype, without cross-influence. Finally, we compared the firing rates between subtypes with the Izhikevich biological neuron models (Huang et al, 2022), which are fitted to neuroscientific experimental data. Given the longer simulation steps of biological neurons compared to aLIF models, we normalized the firing rates of both neuronal models to the range [0.1, 1.0] and then calculated the Pearson correlation coefficient.
>
> **Normalization is not an additional technique but a straightforward formula:** $0.1+0.9 \times \frac{x-min}{max-min}$, where $min$ and $max$ represent the highest and lowest firing rates among 13 neural subtypes, respectively. The subtypes 6, 7, and 9 you mentioned are inhibitory neurons, with subtypes 6 and 7 being the most frequently discharging Parvalbumin (PV) neurons. Subtypes 1, 2, and 5 represent three distinct excitatory neuronal populations. It can be observed that in our trained model, the firing rate of PV subtypes is the highest, while the overall firing rate of excitatory neurons is comparatively lower, which is consistent with neuroscientific findings (Petersen et al, 2013).
>
> We have refined the content in Section 4.3 to make above experimental process clearer.
>
> > W2.3: For the weight degree distribution: one-sided metrics evaluated on variables with low variance will typically also display a longer tailed distribution in networks after training, as the variable gets smudged and the interval is only open to one side. Without a control of sorts it’s hard to understand whether in this case this is an effect of the cortical constrains or simply due to backpropagation and this being a directed graph. Evaluating this or a similar metric for e.g. the LSTM could possibly serve as a control / comparison here too.
>
> **A7:**  Thank you for your inquiry about the specificity of the degree distribution in our model. **We have performed additional experiments on the SRNN 256 model, and the findings reveal that the long-tailed degree distribution is indeed a distinctive feature of our model.**
>
> I would like to first clarify that degree distribution is an analytical method within the field of graph theory, rather than a technique applied to conventional ANNs or SNNs. Our barrel cortex model can be considered a graph due to its faithful replication of the types, quantities, and projection strengths of neurons in brain networks. Consequently, each neuron in our model can be corresponded to a biological neuron in the sensory cortex, allowing us to construct a graph. In contrast, complex and abstract operations like convolutional layers and Transformers **complicate viewing conventional ANNs/SNNs as typical brain networks**, reducing the biological interpretability of degree distribution statistics and increasing technical implementation challenges.
>
> However, the simple SRNN 256 model (Yin et al. 2020), although lacking in biological interpretability, is technically feasible for statistical degree distribution, and thus we have supplemented comparative experimental results for it (**see Figure 6 in the supplementary material**). The results indicate that the degree distribution of the SRNN 256 model is quite uniform before and after training, without any significant long-tail effect.

---

> ### Author Response · Authors · 2024-11-23
> **Response to Reviewer tE3s (part 4)**
>
> This is a continued response to the remaining weaknesses and questions.
> ## Response to Weaknesses
> > W2.4: Regarding the connectivity probability: this is more convincing, yet only IFF the network was properly re-initialized for the changed connectivity, e.g. using the proposed coefficient of variation technique.
>
> **A8:**  Thank you for your concern about the reliability of the connectivity probability ablation experiment. Following this important comment, **we have reinitialized the comparative networks depicted in Figure 6B**, with the exception of random connections, by selecting initial parameters that yield higher CV. The initialization procedure is detailed **in Figure 7 in the supplementary material**. The training outcomes of the reinitialized comparative networks continue to be significantly weaker than those with anatomical connections, aligning with our previous conclusions.
>
> > W2.5: However, it probably cannot be expected to also get explanations as to “why” we see these effects in a single paper, and therefore an extension of the future work section and making the readers aware of these possible limitation should be enough to address most of these issues.
>
> **A9:** We have extended the revised conclusion section with additional content regarding this matter, encompassing the biologically constrained initialization and more detailed network state analysis.
>
> ## Response to Questions
> > Q1: The firing activity in figure 2D seem very synchronized and periodic, does this reflect real work data, and do you have example visualization of such data?
>
> **A10:** We have provided a detailed response to this question in our reply to W 1.3 (A3 in part1).
>
> > Q2: Any explanation as to why the anatomic connection probability is best? Were the initialization adjusted individually for each of the connection probabilities?
>
> **A11:** As responded to W 2.4 in A8, we have supplemented the initialization for other connection probabilities, with the process detailed in **Figure 7 in the supplementary material**, and the new results are consistent with the previous findings.
>
> We believe that the reason anatomical connection probabilities are advantageous is that the brain's structure has been optimized through the process of biological evolution and is encoded in the genome ("Probabilistic skeletons endow brain-like neural networks with innate computing capabilities," 2021), which sets it apart from other connection probabilities. Specifically, the diversity of neurons and their anatomical connections together form a stable network state, and disruptions to these connection probabilities destabilize this network state.
>
> > Q3: For each of the network characteristics findings; do you believe you find this effect due to the biological constrains or simply gradient based training in networks?
>
> **A12:** We conducted comparative experiments on selective firing and degree distribution in other SNNs, and the results indicate that these biological characteristics are unique to our model. **For specifics, refer to the responses to W 2.1 and W 2.3 (A5 and A7)**, respectively.
>
> > Q4: Where any normalization or regularization techniques used for training, or gradient clipping?
>
> **A13:** We smoothed the training process by increasing the rate of learning rate decay, thus no additional gradient regularization techniques were introduced. Details are described in our response to W1.2 (A2 in part1).

---

> > ### Comment · Reviewer_tE3s · 2024-11-24
> > **Some methodological concerns remain and contributions still not well aligned.**
> >
> > Thank you for your detailed explanations and additional experiments and corrections, although quite late in the discussion period.
> >
> > **A1:** The presented methods are not training “biological neural networks”, but maybe biologically inspired / constrained networks. Training such networks is not binary successful Yes / No, as you yourself show in Figure 2C by evaluating Accuracy (continuous).
> >
> > To further highlight the problem in this case, see you current formulation of contribution;
> >
> > “Efficient training framework: Our novel training algorithm is specifically tailored to accommodate the diversity of neuronal types and their anatomically based projection strengths within the network. This includes the introduction of an order parameter to measure the synchrony of neuronal activity, thereby identifying a trainable initial state. This framework is not only applicable to the barrel cortex but can also be extended to other biologically constrained models.”
> >
> > Notice how none of the three points you mentioned to justify “efficiency” in your A1 answer are mentioned (binary success of training, little required GPU resources, generalizability of results).
> >
> > Beside the self-inconsistency, I don't believe that “efficiency” is applied to describe “success of training” or “generalizability of results” in the related literature. However, I would be open to example references and literature surveys that in the context of biologically inspired / constrained networks employ this term in this way to change my mind on this.
> >
> > **A2:** The accuracy plot looks more reasonable now, however, still large jumps remain.
> > **Q1:** What does the training accuracy history look like?
> >
> > **A3:** It is good to see that the trained network displays more plausible spiking patterns, however, considering that the initialization is pitched as a core contribution, seeing the initialized network display pathological fighting patterns is still concerning. In supplementary Figure 1 both axes are missing a unit like “ms” or “Hz” respectively, so it is quite hard to see whether these are plausible ranges.
> >
> > **A4:** If I’m not mistaken, the mentioned reasons are not explaining why a typical synapse scale of ‘’’~1/sqrt(in_deggree)’’’ could not be applied (can be also done for only positive weighs, weights would still be small, and an additional global gain parameter can still be incorporated). Also note that according to your Figure 1 the network is still initialized in a pathological regime, which rather implies a bad initialization.
> >
> > **A5:** Thank you for the additional analysis, the firing selectivity experiment is more convincing now.
> >
> > **A6:** This is still not convincing. If I’m not mistaken, the Pearson correlation coefficient is scale free, so re-normalizing should not be necessary if say due to the different simulation technique one yields 2x the firing rate.
> > **Q2:** Why is there an offset in the normalization? What is the correlation coefficient and p-value without the normalization technique applied? Can you show the corresponding plot?
> >
> > Also, if a biological significance of the resulting parameter settings is claimed, a more convincing experiment could be to re-initialize the specific neuron subtypes to their corresponding mean parameter values of the first training run.
> > **Q3:** Would we observe a drift in mean again, or would the mean not change during training? Latter would indicate some more fundamental finding.
> >
> > **A7:** Thank you for the additional weight degree distribution analysis, this better contextualizes the previous results. However, I cannot follow the explanation as to why this analysis couldn’t be done on an e.g. LSTM: it has a well established interpretation as a recurrent layer of neurons, albeit with positive and negative synapses. However, this was also the case for the SRNN.
> >
> > **A8:** Thank you for the additional experiment, this is more convincing now.
> >
> > **A9:** I appreciate the additions of limitations (although quite short) and future work (although seemingly biased by my previous comment; I would rather value your perspective on the matter).
> >
> > **Some key remaining concerns are:**
> >
> > 1) Efficiency of training still claimed and not convincingly shown (efficacy more plausible).
> > 2) The pathological firing pattern of the initialized networks (implications for trainability, impact on network statistics before/after training).
> > 3) Interpretability of the differentiated neurons / firing rate statistics (and applied normalization techniques).
> >
> > Finally, and most crucially, **the claimed contributions still don’t align well with the actual shown contributions**, or remain exaggerated. The title, abstract and contributions as well as conclusions should be adjusted accordingly. Something along the lines of:
> >
> > A BIOLOGICALLY CONSTRAINED MODEL OF THE MOUSE BARREL CORTEX CAN EFFECTIVELY INTEGRATE WHISKER SENSORY INFORMATION AND REPLICATES KEY BIOLOGICAL NETWORK STATISTICS.

---

> > > ### Author Response · Authors · 2024-11-25
> > > **Response to Reviewer tE3s' Follow-up Comments （part 2）**
> > >
> > > This is the response to the remaining comments 4-6.
> > > ## Response to minor comments
> > > > 4. The presented methods are not training “biological neural networks”, but maybe biologically inspired / constrained networks. Training such networks is not binary successful Yes / No, as you yourself show in Figure 2C by evaluating Accuracy (continuous).
> > > To further highlight the problem in this case, see you current formulation of contribution;
> > > “Efficient training framework: Our novel training algorithm is specifically tailored to accommodate the diversity of neuronal types and their anatomically based projection strengths within the network. This includes the introduction of an order parameter to measure the synchrony of neuronal activity, thereby identifying a trainable initial state. This framework is not only applicable to the barrel cortex but can also be extended to other biologically constrained models.”
> > > Notice how none of the three points you mentioned to justify “efficiency” in your A1 answer are mentioned (binary success of training, little required GPU resources, generalizability of results).
> > > Beside the self-inconsistency, I don't believe that “efficiency” is applied to describe “success of training” or “generalizability of results” in the related literature. However, I would be open to example references and literature surveys that in the context of biologically inspired / constrained networks employ this term in this way to change my mind on this.
> > >
> > > **R4:** We appreciate your clarification regarding the distinction between biological neural networks and biologically constrained models. We agree with your differentiation and affirm that our model belongs to biologically constrained models, not biological neural networks. We have carefully reviewed the manuscript to ensure that this confusion is avoided. As we responded in R1, we have already revised the title of the paper and other relevant content to better align with our contributions.
> > >
> > > > 5. The accuracy plot looks more reasonable now, however, still large jumps remain. Q1: What does the training accuracy history look like?
> > >
> > > **R5:** Thank you for your question regarding the training process. We have provided the training accuracy **in Figure 9 of the supplementary material**, which aligns with the changes in test accuracy shown in Figure 2C. Further, we have illustrated the changes in gradient norms throughout the training process and tested the impact of several gradient clipping methods on training. The results show that they did not significantly improve the stability of training. Despite this, we agree with your concerns about training stability. Exploring gradient constraints or other training methods (e.g. STDP) for biologically constrained models is indeed valuable.
> > >
> > > > 6. It is good to see that the trained network displays more plausible spiking patterns, however, considering that the initialization is pitched as a core contribution, seeing the initialized network display pathological fighting patterns is still concerning. In supplementary Figure 1 both axes are missing a unit like “ms” or “Hz” respectively, so it is quite hard to see whether these are plausible ranges.
> > >
> > > **R6**: We appreciate your question regarding the neuron activity, and we have provided a detailed explanation of synchronous discharges in R2. Following your suggestion, **we have included units in Figure 1 of the supplementary material**. We would like to clarify that the entire model is a simulation conducted on silicon, and the whisker sweep dataset is also a simulated task. This could potentially lead to a lack of alignment between the absolute values of the metrics and those from biological experiments. Our primary interest lies in the gradients and relative relationships among the subtypes of neurons.

---

> > > > ### Comment · Reviewer_tE3s · 2024-11-25
> > > > **Final suggestions, in particular wrt contribution.**
> > > >
> > > > Thank you for the clarifications, additional plots, modified plots, and adjustment of contributions in the respective sections.
> > > >
> > > > **Regarding the firing initialization:** the given explanation as to “why” we see this firing pattern at initialization is sound, however, this doesn’t make it an intuitively good initialization, especially considering the trained networks firing pattern. It seems to be more like a limitation of the proposed initialization.
> > > >
> > > > **Regarding realism of network:** since you noted that “The goal of our model is not to simulate the neural activity in the brain as **realistically** as possible, but to strike a balance between biological authenticity and behavioral functionality” and the initialization is far from realistic, having “Biologically **realistic** network construction” is a mismatch.
> > > >
> > > > **Regarding training framework:** if I understand correctly, the core novelty is the “order parameter” based initialization. If this is the only thing (not counting the network itself toward the framework) this doesn’t justify calling it a training framework (a comprehensive approach tackling multiple areas of the optimization process) or training algorithm (typically would be a novel optimizer or scheduler), but more an “initialization technique”. Otherwise:
> > > > Q: How is the contribution of “training framework” justified, as opposed to e.g. initialization technique?
> > > >
> > > > I would like to make three final suggestions:
> > > > 1) include the pathological firing init in some form in the limitations
> > > > 2) use a more appropriate / concrete word than “realistic” to refer to the proposed network (contribution)
> > > > 3) reformulate the contribution of “training framework” to something more appropriate / concrete
> > > >
> > > > The changes should be again reflected in the abstract, introduction and conclusion.

---

> > > > > ### Author Response · Authors · 2024-11-26
> > > > > **Response to the revision of contributions**
> > > > >
> > > > > We are grateful for your feedback regarding our manuscript. In response, we have refined the relevant content to accurately represent the true contributions.
> > > > > ## Response to suggestions
> > > > > > Suggestion 1: nclude the pathological firing init in some form in the limitations
> > > > >
> > > > > **Response:**  We have explicitly discussed the limitations of synchronous firing in the conclusion section, **as follows:** "Figure 1 in the supplementary material presents raster plots of our model, where abnormal synchronous firing (with a higher CV score indicating stronger intensity) can still be observed in the initial network. This is due to all neurons being assigned the same parameter values at the start. Although this does not affect the training and disappears in the trained model, it reduces the biological authenticity of the initial network. Further constraining neuronal activity can bring it closer to biologically realistic networks."
> > > > >
> > > > > > Suggestion 2: use a more appropriate / concrete word than “realistic” to refer to the proposed network (contribution)
> > > > >
> > > > > **Response:** We have unified the description of our model throughout the manuscript by using the term "biologically constrained model".
> > > > >
> > > > > > Suggestion 3: reformulate the contribution of “training framework” to something more appropriate / concrete
> > > > >
> > > > > **Response:** We have replaced the term "training framework" with "pipeline for model construction and training". Now, we clarify our manuscript's contributions as follows: 1. Biologically constrained barrel cortex model. 2. Pipeline for model construction and training. 3. Establishing a novel spiking whisker sweep dataset.
> > > > >
> > > > > We have reflected the aforementioned revisions throughout the entire manuscript.

---

> > > > > > ### Comment · Reviewer_tE3s · 2024-11-26
> > > > > > **Adjusting my rating.**
> > > > > >
> > > > > > Thank you for addressing the remaining main concerns.
> > > > > >
> > > > > > 1) The claimed contributions (and description in abstract, introduction and conclusion) are now more concrete and better aligned with what is being shown, and the readers are made aware of the limitations of the work.
> > > > > > 2) The network statistics findings have been supplemented with ablations / analysis of other networks for comparison, making the findings more interpretable / tangible and likely ruling random drifts as the primary reason out.
> > > > > >
> > > > > > Therefore I will increase the soundness score to 3 and the overall score to 8.

---

> > > ### Author Response · Authors · 2024-11-25
> > > **Response to Reviewer tE3s' Follow-up Comments （part 3）**
> > >
> > > This is the response to the remaining comments.
> > > ## Response minor comments
> > > > 7. If I’m not mistaken, the mentioned reasons are not explaining why a typical synapse scale of ‘’’~1/sqrt(in_deggree)’’’ could not be applied (can be also done for only positive weighs, weights would still be small, and an additional global gain parameter can still be incorporated). Also note that according to your Figure 1 the network is still initialized in a pathological regime, which rather implies a bad initialization.
> > >
> > > **R7:** We appreciate your question regarding initialization, and we believe synaptic scaling can indeed be integrated into our method. As previously mentioned, we aim to enhance the generalizability of our method by training biologically constrained models using simple techniques, hence the use of the most intuitive uniform distribution for initialization. In fact, any initialization method that satisfies the three conditions in A4 could be tested in our model. We consider synaptic scaling to be a valuable idea as it can limit excessively large synaptic weights and has a certain biological basis (''Astroglia-specific contributions to the regulation of synapses, cognition and behaviour,'' 2020). It is worth considering how to construct biologically plausible and effective synaptic constraints. For instance, the presynaptic current in our model is linearly accumulated, while in the actual cortex, they are integrated nonlinearly. Coupled with your previous mention of membrane potential initialization (Rossbroich et. al 2022), we will continue to explore the optimization of model initialization methods.
> > >
> > > We have provided a detailed explanation of synchronous firing in R2.
> > >
> > > > 8. This is still not convincing. If I’m not mistaken, the Pearson correlation coefficient is scale free, so re-normalizing should not be necessary if say due to the different simulation technique one yields 2x the firing rate. Q2: Why is there an offset in the normalization? What is the correlation coefficient and p-value without the normalization technique applied? Can you show the corresponding plot?
> > >
> > > **R8:** We have responded in detail to this question concerning normalization in R3. Upon testing, we found that the Pearson correlation coefficients are consistent before and after normalization.
> > >
> > > > 9. Also, if a biological significance of the resulting parameter settings is claimed, a more convincing experiment could be to re-initialize the specific neuron subtypes to their corresponding mean parameter values of the first training run. Q3: Would we observe a drift in mean again, or would the mean not change during training? Latter would indicate some more fundamental finding.
> > >
> > > **R9:** Thank you for your novel idea regarding the gradient of neuronal subtype dynamics. Of course, if the network state changes, the dynamic parameters would differentiate again. In fact, **we previously conducted such an experiment**, assuming that reinitializing with trained neuronal subtype parameters would be more appropriate, but the results did not show significant improvement. We believe there are two reasons for this: **1.** The gradient of neuronal dynamics does not function in isolation, but constitutes the entire dynamic system together with the synaptic weights after training. **2.** As shown in Figure 5A, the mean is only a general indicator. In reality, the parameter values of many individual neurons cluster around the mean. Assigning the same mean value to all neurons is a fundamentally different state from each neuron having its own state that only equals the overall mean.
> > >
> > > > 10. Thank you for the additional weight degree distribution analysis, this better contextualizes the previous results. However, I cannot follow the explanation as to why this analysis couldn’t be done on an e.g. LSTM: it has a well established interpretation as a recurrent layer of neurons, albeit with positive and negative synapses. However, this was also the case for the SRNN.
> > >
> > > **R10:** Thank you for your question about the comparison models. We prioritized comparing with SNNs because they are more similar to our model than ANNs. Following the comment, **we added the degree distribution of a LSTM model in Figure 6 of the supplementary material**. The additional results did not show a significant long-tail effect, consistent with our previous conclusions.

---

> ### Author Response · Authors · 2024-11-25
> **Response to Reviewer tE3s' Follow-up Comments （part 1）**
>
> We appreciate your thorough and constructive review of our work. It took us some time to conduct additional experiments for all reviewers, hence the delayed response, but we will strive to answer all the issues you have raised. Please see our detailed responses below.
> ## Response to key comments
> > 1. Efficiency of training still claimed and not convincingly shown (efficacy more plausible).
>
> **R1:** We appreciate your insights on refining our contribution. In order to better articulate the value of our work, **we have followed your advice and revised the title, abstract, and introduction section**. The current title of the manuscript is: " Effective Integration of Whisker Sensory Information and Replication of Key Biological Network Statistics by a Biologically Constrained Model of the Mouse Barrel Cortex" The term "Efficient" from the original manuscript has been removed to avoid any ambiguity.
>
> > 2. The pathological firing pattern of the initialized networks (implications for trainability, impact on network statistics before/after training).
>
> **R2:** Thank you for your question regarding synchronous firing in the initial model. We believe that **synchronous firing can be primarily understood from three aspects:** **1.** In the initialization phase, all neurons are assigned the same initial values, causing them to fire in unison when subjected to strong external inputs. **2.** The aLIF model is more prone to firing than biological neuron models, such as the HH model, making it easier for simultaneous firing to propagate among neuronal subtypes. **3.** Once activated, inhibitory neurons suppress the firing of neurons in the model. Simultaneously, the spike-induced threshold increment (see Equation 3) constrains further neuron firing, thus forming a silent period after concentrated firing.
>
> However, this demonstrates that the initial network does not sufficiently mimic cortical circuits, thereby emphasizing the significance of adjusting the internal state of the model with biological constraints during the learning of behavioral functions. **The goal of our model is not to simulate the neural activity in the brain as realistically as possible, but to strike a balance between biological authenticity and behavioral functionality**. Detailed biological neural network models, such as the "Reconstruction and Simulation of Neocortical Microcircuitry" (2015), are currently untrainable and lack behavioral functionality.
>
> Regarding the CV measure, we have provided raster plots of the initial network with different CV scores (**see the revised Figure 1 in the supplementary material**). As can be seen, although there is synchronous firing, there are differences in the degree. In general, we hope to find such an initial state through the CV measure: neurons neither remain silent forever nor fire excessively due to minimal input, but are in a higher state of freedom (as can be seen, there are many irregular spikes between concentrated firings). This is a general principle, not a precise derivation.
>
> We agree with your assessment of the importance of bio-plausible network states. Assigning differentiated parameters to neuronal subtypes at the outset, based on neurophysiological data (e.g., the Allen Cell Type Database), is a research direction worth considering. In addition, a detailed analysis of the network's internal state also relies on the inspiration from more brain dynamics results (e.g., Tian et al. 2022).
>
> > 3. Interpretability of the differentiated neurons / firing rate statistics (and applied normalization techniques).
>
> **R3:** Thank you for your inquiry regarding our process of calculating firing rate similarity. This normalization indeed does not affect the calculation of the Pearson correlation coefficient. We employed normalization merely to conveniently visualize the two types of firing rates within a unified range. The addition of a 0.1 offset is also for better visualization; we do not want the subtype with the lowest firing rate to appear blank in the bar graph of Figure 5B.
>
> Meanwhile, we have clarified in Section 4.3 of the main text that this **normalization not related to the Pearson correlation calculation, but is solely for better visualization**.

---

### Official Review · Reviewer_PD9T · 2024-11-02

**Soundness:** 3
**Presentation:** 3
**Contribution:** 2
**Rating:** 6
**Confidence:** 3

**Summary:**

This paper developed a trainable Barrel circuit, whose connection backbone is constrained by biological data of Barrel cortex, to classify object based on (simulated) whisker input. The network is composed of identical adaptative LIF neurons, with tranable timescale parameters. The parameter distribution and response property are analyzed after the training, which is argued to have qualitative similiarity with biological data. The model is compared with several baseline ANN and SNN models based on spiking-based dataset.

**Strengths:**

This paper build up a trainable bioplausible model for touch, combining the function with bio-interpretation. This is an important and interesting direction in general. On the techinical side, the paper applies the coefficient of variation to determine the desirable initial configuration of a dynamical network model. It also convert a touch dataset into spiking version by Bernoulli sampling based on firing rate. On the discovery side, the author find bio-related feature in the network after the training, including neural selectivity, shift in timescale parameters, similiar response activity as biological data, long-tailed degree distribution, etc. These make it a promissing machine learning approach to study touch.

**Weaknesses:**

While the overall picture is promissing, the novelty and significance of the paper may not meet the standard of ICLR.
- The training of aLIF network has been realized in a list of previous works, e.g. Chen2022 cited in the paper. And using connection probability to constrain the network topology is also not new
- The proposed CV measure is only showcased in Fig2 based on three selected configurations. It is not clear whether the CV is a significant/convincing measure of the initialization for training and whether it is generalizable to other dynamical bio-plausible models. For example, it is not clear how three configuration is selected and the results is not averged by different random seeds. Also, in FigC, the different initialization do not affact the initial traning process(0-10 epoch), which does not make sense to me if three initialization actually results in bifurcation of the dynamics regions. And the Fig.2 D all shows global synchrony dynamics (even for high CV), which still lack of variability and is not a realistic working regime in the brain .
-  The conversion of real-valued dataset to spiking-based dataset is not a significant contribution, and it is not clear whether the converted spiking data actually mimic the input in biological circuit.
- As a result, while the overall picture is interesting, the specific technical contribution of this paper beyond the related work is limited.

On the experiment side, the significance of result is also my concern:
- Compare with ANN: the ANN is compared based on spiking dataset. So it is not clear whether the poor performance of ANN is due to the data type of input. Since it is not necessary to use spiking data as input to spiking network, the author should also compare the model with ANN on original non-spiking dataset, to disentangle the whether it is the network dynamic or just the data type that acturally make the difference.
- Firing selectivity: it is not a suprise to see the symmetry breaking after the training. The author should compare the selectivity property with other SNN models (after training on this task) to stress the significance of the result.
- The shift of parameter is also not a suprise. It would be a suprise if the shifted distribution mimic that in real circuit. Without such direct comparison, it is not clear whether the shift is a specific property of this model, that have a meaning or just a general phenomenon due to training, given heterogeneous connection backbone.
- The firing rate profile, it is not clear how the this profile is computed: is identical curent applied to each subpopulation seperately or applied to the entire network? Besides, the profiles of different neuron type differ not only in firing rate but also specific spike patterns, like tonic, bursting,...and their responsive property to stimulus ('on','off','on-off'). Can author show more detailed comparison with biological response profile as support.
- For the long-tail degree distribution, it is also not clear whether it is a specific property of this model or a general property of training. I suggest the author to compare the result with other SNN / ANN, to clarify this.

Lastly, the code of the model is not provided, so it is hard to evaluate the solidness of the results.

**Questions:**

See above.

---

> ### Author Response · Authors · 2024-11-24
> **Response to Reviewer PD9T (part 1)**
>
> Thank you for taking the time to review. Please see our detailed response to the issues you have raised.
> ## Response to Weaknesses
> > W1: The training of aLIF network has been realized in a list of previous works, e.g. Chen2022 cited in the paper. And using connection probability to constrain the network topology is also not new
>
> **A1:**  Thank you for your critique of the innovation in our work; however, we respectfully disagree with your viewpoint. Our manuscript does not claim contributions to aLIF or connectivity probabilities—**our genuine contribution is the development of an novel barrel cortex model that balances biological plausibility with sensory function**.
>
> Our model features the following two innovations: **1.** It faithfully replicates the anatomical structure of the barrel cortex, and is capable of being trained on a sensory function task, thereby possessing both the structural authenticity of sensory cortex and its behavioral functionality. **2.** During the learning process of the sensory task, our model spontaneously aligns with certain characteristics of real cortex, which are not observed in conventional ANNs/SNNs (we have conducted supplementary comparative experiments as per your comments).
>
> **There are at least two significant differences between our work and Chen et al., 2022 you mentioned:** **1.** Chen et al.'s work is predicated on modeling the V1 brain region, while we focus on the barrel cortex. This implies that their model may not be adept at studying sensory functions. **2.** As stated in their paper, Chen et al.'s model requires 60 hours of training on 160 A100 GPUs each session. In contrast, our work solely relies on a single A100 GPU, rendering it more suitable for most laboratories.
>
> **Regarding the aLIF model and connection probability you mentioned, our motivations and implementations also significantly diverge from existing works.** While previous aLIF models have been used to bolster the training stability of traditional SNNs (Yin et al., 2020), we employ them to distinguish the dynamic gradients of neuronal subtypes, incorporating models the types of pre-synaptic currents (excitatory or inhibitory, see Equation 4). Besides, existing works typically employ connection probability to limit the number of model parameters (e.g., Zheng et al., 2024), but we use it to mimic anatomically-based neuronal projection strengths. We strive to use concise techniques to build biologically realistic models because **we believe that simpler methods offer stronger generalizability, hence making them more adaptable to other neural circuit studies**.
>
> In summary, our primary contribution is a novel barrel cortex model, not minor techniques. **To the best of our knowledge, there is currently no existing barrel cortex model that successfully replicates anatomical structures while also possessing sensory functions**.

---

> ### Author Response · Authors · 2024-11-24
> **Response to Reviewer PD9T (part 2)**
>
> This is the continued response to the remaining weaknesses 2-3.
> ## Response to Weaknesses
> > W2: The proposed CV measure is only showcased in Fig2 based on three selected configurations. It is not clear whether the CV is a significant/convincing measure of the initialization for training and whether it is generalizable to other dynamical bio-plausible models. For example, it is not clear how three configuration is selected and the results is not averged by different random seeds. Also, in FigC, the different initialization do not affact the initial traning process(0-10 epoch), which does not make sense to me if three initialization actually results in bifurcation of the dynamics regions. And the Fig.2 D all shows global synchrony dynamics (even for high CV), which still lack of variability and is not a realistic working regime in the brain.
>
> **A2:** Thank you for raising questions about the persuasiveness of the CV measure. The sensitivity of biological neural networks to minute parameter changes, resulting in alterations to their internal states, is a characteristic feature that also complicates their training. Therefore, we have adopted the CV measure from studies on brain critical states (Tian et al. 2022) to quantify internal states of our network. Its primary goal is to find an initial state in which a biological neural network can be trained. As we mentioned in the response to W1, we believe that the simpler and more direct the metric, the better its generalizability, making it easier to apply to other neural circuits.
>
> Furthermore, **we present a more comprehensive CV heatmap in Figure 2B**, which more clearly displays the evolution of the network state with respect to initial parameters. The three initial settings in Fig 2C represent samples we selected from different CV regions, emphasizing the impact of network states on training performance.
>
> Although the initial phase of training for all three initialization configurations shows an increase in classification accuracy, this does not imply that they share similar states (akin to different directions in optimization). Therefore, we select the model based on the best accuracy achieved during the network training process.
>
> Regarding the initial synchronous firing you mentioned, this occurs because all aLIF neurons are assigned the same initial parameters. Additionally, the aLIF model is more prone to firing compared to biological neurons, such as the Hodgkin-Huxley model, which reflects a common distinction between spiking neural networks and biological neural networks.
>
> To better address your question about synchronous firing, **we have additionally presented the raster plot of the network after training (see Figure 1 in the supplementary material)**. It can be observed that pathological synchronous firing disappears in the trained network, and the relative firing rates of excitatory and inhibitory subtypes correspond with phenomena observed in actual circuits ("Synaptic Computation and Sensory Processing in Neocortical Layer 2/3," 2013, and "A Cellular Resolution Map of Barrel Cortex Activity during Tactile Behavior," 2015). This indicates that our model can adjust its internal state through learning sensory tasks, demonstrating the research value of enabling existing biological neural network models to undergo learning and training.
>
> > W3: The conversion of real-valued dataset to spiking-based dataset is not a significant contribution, and it is not clear whether the converted spiking data actually mimic the input in biological circuit.
>
> **A3:** Thank you for your questions regarding the contribution of our spiking-based dataset. We would like to firstly clarify that biological neurons in the brain cortex communicate through spikes, and **we adhere to this fundamental principle to ensure the biological authenticity of our barrel cortex model**.
>
> Secondly, it is nearly impossible to precisely record the spiking trains of individual neurons during mouse sensory-behavioral experiments using existing techniques. This also results in a lack of datasets, hindering the training of biological neural networks. **The goal of our work is not to simulate brain signals as realistically as possible, but to propose an approximate method to enable the training of biologically constrained models.** Therefore, we established the simulated spiking whisker sweep dataset referencing the rate coding of sensory signals by the thalamus neurons (Lee et al, 2024), which can serve as a testing platform for subsequent models related to the sensory cortex.
>
> Additionally, we would like to elaborate that transforming real-valued datasets into spike form is a research direction in neuromorphic computing, which aids in facilitating benchmarking for neuromorphic software and hardware (e.g. "The Heidelberg Spiking Data Sets for the Systematic Evaluation of Spiking Neural Networks," 2022). The contribution of our dataset should also be considered within this broader field.

---

> ### Author Response · Authors · 2024-11-24
> **Response to Reviewer PD9T (part 3)**
>
> This is the continued response to the remaining weaknesses 4-5.
> ## Response to Weaknesses
> > W4: Compare with ANN: the ANN is compared based on spiking dataset. So it is not clear whether the poor performance of ANN is due to the data type of input. Since it is not necessary to use spiking data as input to spiking network, the author should also compare the model with ANN on original non-spiking dataset, to disentangle the whether it is the network dynamic or just the data type that acturally make the difference.
>
> **A4:** Thank you for raising this question regarding the comparison results of ANNs. The original manuscript indeed presents a discrepancy in the input forms between ANNs and SNNs, which could lead to divergent interpretations of the results. To address this, **we have conducted supplementary tests to evaluate the classification performance of four SNNs on the real-valued whisker sweep dataset**. The results have been appended to the last row of Table 1 and alson showed below:
>
> | ANNs | ST CNN | DB LSTM | RNN+ | UGRNN |
> |---|---|---|---|---|
> | Real-valued | 67.2 | 70.9 | 78.2 | 76.4 |
>
> | SNNs | SRNN 256 | STSC SNN | TA SNN | Ours |
> |---|---|---|---|---|
> | Spiking-based | **81.8** | 80.0 | **81.8** | **81.8** |
> | Real-valued | 85.5 | 87.3 | 83.6 | **89.1** |
>
> Additional results demonstrate a significant improvement in the performance of all SNNs when processing the real-valued inputs  compared to spike-based inputs, with our barrel model continuing to outperform the others.
>
> The enhancement in SNNs' performance with real-valued inputs is easily explicable. As detailed in Section 3.3, we employ random sampling to generate independent spikes that describe the real value at each moment, effectively introducing perturbations. Furthermore, expanding the time steps of the original dataset from 110 to 550 is less favorable for the firing of spiking neurons within the network. In conclusion, the additional results further confirm the comparative performance of our barrel model among mainstream computational models.
>
> > W5: Firing selectivity: it is not a suprise to see the symmetry breaking after the training. The author should compare the selectivity property with other SNN models (after training on this task) to stress the significance of the result.
>
> **A5:** Thank you for the crucial suggestion to compare firing selectivity with other SNNs. **We have conducted analogous experiments on three other SNNs, and the results indicate that this is a distinctive feature of our model**.
>
> Following the important comment, we have supplemented our study with experiments on the selective firing of other three SNNs mentioned in the manuscript (**see Figure 3 in the supplementary material**). It can be observed that both the SRNN 256 and TA SNN do not exhibit significant firing selectivity before and after training, indicating that indicating that the firing selectivity is not a universal phenomenon.
>
> **There are at least three significant differences compared to our model, despite numerous neurons in the STSC SNN demonstrated preferred firing:** **1.** Our model did not have significant firing selectivity in its initial state; the preferential firing emerged spontaneously during the learning of a sensory function. In contrast, approximately half of the neurons in the STSC SNN showed selectivity from the outset. **2.** The firing preferences in our model are more concentrated, forming distinct functional clusters, whereas the STSC SNN shows a more uniform distribution. **3.** Fundamentally, each neuron in our model can be correlated with a biological neuron in the cortex, offering the potential for further comparison with actual neural circuits, a feature that conventional SNNs lack in terms of biological interpretability.

---

> ### Author Response · Authors · 2024-11-24
> **Response to Reviewer PD9T (part 4)**
>
> This is the continued response to the remaining weaknesses 6-7.
> ## Response to Weaknesses
> > W6: The shift of parameter is also not a suprise. It would be a suprise if the shifted distribution mimic that in real circuit. Without such direct comparison, it is not clear whether the shift is a specific property of this model, that have a meaning or just a general phenomenon due to training, given heterogeneous connection backbone.
>
> **A6:** We exactly employ the comparison of neuronal subtype firing rates depicted in Figure 5B to assess the similarity between the dynamic shift in our trained model and those observed in the real cortex. The results show that our model has risen from no correlation before training to a Pearson correlation coefficient of 0.66 after training. The specific methodology will be detailed in the response to the next Weakness (A7).
>
> > W7: The firing rate profile, it is not clear how the this profile is computed: is identical curent applied to each subpopulation seperately or applied to the entire network? Besides, the profiles of different neuron type differ not only in firing rate but also specific spike patterns, like tonic, bursting,...and their responsive property to stimulus ('on','off','on-off'). Can author show more detailed comparison with biological response profile as support.
>
> **A7:**  Thank you for your question regarding the clarity of our firing rate comparison, below is a detailed explanation of the implementation methodology for Figure 5B. Initially, each neuron in our barrel cortex model was assigned identical initial parameters, therefore a consistent response to identical input currents (**red reference line in Figure 5B**). After training, we preserved the differentiated parameters for each neuronal subtype within our network. Subsequently, a sustained 1mV external current was applied to each neuron within the subtypes for a duration of 550ms, and the total spikes count for the entire subtype was tallied. **This was conducted in isolation for each subtype, without cross-influence**. Finally, we compared the firing rates between subtypes with the Izhikevich biological neuron models (Huang et al, 2022), which are fitted to neuroscientific experimental data.
>
> Given the longer simulation timestep of biological neurons compared to the aLIF model, we normalized the firing rates of both neuronal models to the range [0.1, 1.0] and then calculated the Pearson correlation coefficient. The simple formula for normalization is: $0.1+0.9 \times \frac{x-min}{max-min}$, where $min$ and $max$ represent the highest and lowest firing rates among 13 neural subtypes, respectively. It is important to note that **this normalization does not affect the calculation of the Pearson correlation coefficient**.
>
> Regarding the specific features such as "bursting" that you mentioned, these are currently only observable in detailed biological neural network models (e.g., "Reconstruction and Simulation of Neocortical Microcircuitry," 2015). However, these sophisticated biological models are not amenable to learning and training, and thus lack advanced perceptual functions. Our model aims to strike a balance between biological plausibility and perceptual capabilities. Nevertheless, as stated in our response (A2) to W2, some general similarities with the actual sensory cortex can be observed. For instance, it is evident that inhibitory neurons are more active than excitatory neurons, which aligns with previous studies on the excitatory-inhibitory balance in the barrel cortex (**see Figure 1 in the supplementary material**).
>
> We have refined the content in Section 4.3 to make above experimental process clearer.

---

> ### Author Response · Authors · 2024-11-24
> **Response to Reviewer PD9T (part 5)**
>
> This is the continued response to the remaining weaknesses 8-9.
> ## Response to Weaknesses
> > W8: For the long-tail degree distribution, it is also not clear whether it is a specific property of this model or a general property of training. I suggest the author to compare the result with other SNN / ANN, to clarify this.
>
> **A8:** Thank you for your inquiry about the specificity of the degree distribution in our model. **We have performed additional experiments on the SRNN 256 model, and the findings reveal that the long-tailed degree distribution is indeed a distinctive feature of our model.**
>
> I would like to first clarify that degree distribution is an analytical method within the field of graph theory, rather than a technique applied to conventional ANNs or SNNs. Our barrel cortex model can be considered a graph due to its faithful replication of the types, quantities, and projection strengths of neurons in brain networks. Consequently, each neuron in our model can be corresponded to a biological neuron in the sensory cortex, allowing us to compute the adjacency matrix and construct a graph. In contrast, complex and abstract operations like convolutional layers and Transformers **complicate viewing conventional ANNs/SNNs as typical brain networks**, reducing the biological interpretability of degree distribution statistics and increasing technical implementation challenges.
>
> However, the simple SRNN 256 model (Yin et al. 2020), although lacking in biological interpretability, is technically feasible for statistical degree distribution, and thus we have supplemented comparative experimental results for it (**see Figure 6 in the supplementary material**). Since the SRNN 256 model is initialized with the xavier_uniform method for synaptic weights, we aggregated the absolute values of the synapses weights connected to each neuron as the weighted degree. The results, as shown in the figures below, indicate that the degree distribution of the SRNN_256 model is quite uniform before and after training, without any significant long-tail effect.
>
> > W9: Lastly, the code of the model is not provided, so it is hard to evaluate the solidness of the results.
>
> **A9: We have committed in the abstract to open-source the code immediately upon acceptance of the manuscript.**

---

> ### Author Response · Authors · 2024-11-26
> **Supplement to the manuscript revision**
>
> In order to more clearly align our contributions within the manuscript, following your comments, **we have reformulated the manuscript's title, contributions, and limitations sections**.
>
> The title of the manuscript is now: "Effective Integration of Whisker Sensory Information and Replication of Key Biological Network Statistics by a Biologically Constrained Model of Mouse Barrel Cortex". We have reformulated the contributions to: 1. Biologically constrained barrel cortex model, 2. Pipeline for model construction and training, 3. Establishing a novel spiking whisker sweep dataset. Concurrently, we have discussed the limitations of the current initialization methods and potential future work in the conclusion section.
>
> These changes have been reflected throughout the entire manuscript.

---

> ### Author Response · Authors · 2024-11-28
> **Request for Continued Responses and Discussion**
>
> With the discussion deadline drawing near, we would greatly appreciate it if you could review our responses and reconsider our work. We stand ready to answer any further questions you may have and are more than willing to provide additional clarification as needed. It would be our privilege to address and resolve any remaining concerns. Thank you again for your time and valuable comments.

---

### Official Review · Reviewer_uNoP · 2024-11-03

**Soundness:** 3
**Presentation:** 3
**Contribution:** 3
**Rating:** 8
**Confidence:** 3

**Summary:**

The following paper investigates the gap between biological realism and behavioural functionality of embodied intelligence to develop an efficient training algorithm. The paper introduces a framework for bridging the gap between realism and functionality based on the barrel cortex, where it covers over three contributions: biologically realistic network construction, efficient training framework and establishing a novel spiking whisker sweep dataset.

**Strengths:**

The paper clearly indicates the gap that exists between the training and biological plausibility of models and current progress in closing this gap.
I really like the idea on how the original weights are constrained to simulate the connections to represent the 13 neuronal subtypes.
Mapping from real to discrete space using a sigmoid function for the dataset is interesting.
Training for synchronisation of neuronal activity is a novel and effective approach, enhancing both stability and biological realism in sensory-motor integration models.

**Weaknesses:**

The paper mentions that it has a better classification accuracy compared to models of CNNs, RNNs, LSTMs and SNN. CNNs, for e.g. work on the spatial structure of the dataset, and thus this should be interesting to see in both the original and currently introduced spiking dataset since spikes deal more on the temporality of the data.
The structure of the paper is lacking (it seems to be going back and forth spontaneously between different sections in my opinion) although the information provided is sufficient. An example is that some of the information mentioned in Figure 2 are better mentioned after an explanation on the methodology. These took quite some time to understand until reaching section 3. Moreover, the task mentioned in contributions on the spiking dataset (page 3) would benefit better if it also uses figure 3 as a supplementary material to explain.

**Questions:**

Figure 1A: The figure description seems to need some clarity onto the labels (1,2,3). It is unclear whether the number defines the region or the pathway.
Figure 2A: From my perspective, the connection probability and indication of excitatory and inhibitory connections of the subtypes are not clear. These two appear to be mixed in the diagram and thus require more clarification in the diagram.
Equation 3: Please elaborate more on what beta defines.
More clarity on the dataset used for table 1 and 2: It is mentioned that it uses a spiking whisker sweep dataset in page 3 but it would be better if it is mentioned in the table descriptions as well for better clarity.
The use of the barrel cortex as a foundation to bridge the gap between efficient training and biological plausibility is well-motivated. The introduction of a biologically constrained model, trained by synchronising neuronal electrical activity, is an interesting approach as well. Given the paper’s focus on an efficient training

---

> ### Author Response · Authors · 2024-11-23
> **Response to Reviewer uNoP**
>
> Thank you for your meticulous review. Below is a detailed response to your comments:
> ## Response to Weaknesses
> > W1: The paper mentions that it has a better classification accuracy compared to models of CNNs, RNNs, LSTMs and SNN. CNNs, for e.g. work on the spatial structure of the dataset, and thus this should be interesting to see in both the original and currently introduced spiking dataset since spikes deal more on the temporality of the data.
>
> **A1:** Thank you for pointing out the differences in training datasets between ANNs and SNNs. Indeed, as you have pointed out, both ANNs and SNNs are technically capable of processing real-valued or spike-based inputs. Given that our focus is on biologically plausible SNNs rather than ANNs, **we have conducted additional tests on the performance of the four SNNs mentioned in the manuscript using the original real-valued dataset,** enabling a more equitable comparison between ANNs and SNNs. The results are appended to the last row of Table 1 and are also showed below:
>
> | ANNs | ST CNN | DB LSTM | RNN+ | UGRNN |
> |---|---|---|---|---|
> | Real-valued | 67.2 | 70.9 | 78.2 | 76.4 |
>
> | SNNs | SRNN 256 | STSC SNN | TA SNN | Ours |
> |---|---|---|---|---|
> | Spiking-based | **81.8** | 80.0 | **81.8** | **81.8** |
> | Real-valued | 85.5 | 87.3 | 83.6 | **89.1** |
>
>
> Supplementary results indicate that all SNNs exhibit superior performance with continuous values compared to spiking inputs, with our barrel model maintaining a performance advantage. This enhancement is readily explicable. As detailed in Section 3.3, we employ random sampling to generate independent spikes that describe the real value at each moment, which is analogous to introducing perturbations. Furthermore, extending the original dataset's time steps from 110 to 550 is also detrimental to the firing of spiking neurons within SNNs. In summary, the supplementary results further confirm the comparative performance of our barrel model among mainstream computational models.
>
> > W2: The structure of the paper is lacking (it seems to be going back and forth spontaneously between different sections in my opinion) although the information provided is sufficient. An example is that some of the information mentioned in Figure 2 are better mentioned after an explanation on the methodology.
>
> **A2:** Thank you for drawing attention to the lack of clarity in the structure of our manuscript. The premature introduction of Section 3.3 within Section 3.2 may have caused some confusion. We have revised Figure 2 caption and clarified the connections between Sections 3.2 and 3.3 to address the confusion you noted.
>
> > W3: Moreover, the task mentioned in contributions on the spiking dataset (page 3) would benefit better if it also uses figure 3 as a supplementary material to explain.
>
> **A3:** Following your advice, we have clearly referenced Figure 3 in the contributions section to enhance understandability.
>
> ## Response to Questions
>
> > Q1: Figure 1A: The figure description seems to need some clarity onto the labels (1,2,3). It is unclear whether the number defines the region or the pathway.
>
> **A4:** (1, 2, 3) correspond to trigeminal ganglion neurons, brainstem neurons, and thalamic neurons, respectively, not brain regions. This clarification has been made in the caption of Figure 1.
>
> > Q2: Figure 2A: From my perspective, the connection probability and indication of excitatory and inhibitory connections of the subtypes are not clear. These two appear to be mixed in the diagram and thus require more clarification in the diagram.
>
> **A5:** We have used the shade of pixel colors to represent the strength of connection probabilities and red-blue borders around pixels to denote excitatory or inhibitory connections, respectively. However, this representation may be ambiguous when combined. Therefore, in the revised version of Figure 2A, we have widened the borders for better distinction.
>
> > Q3: Equation 3: Please elaborate more on what beta defines.
>
> **A6:**  In Equation 3, β represents the coefficient that amplifies or attenuates the threshold increment induced by spikes, and it is set to 1.8 in our paper. We have added an explanation of β in the main text.
>
> > Q4: More clarity on the dataset used for table 1 and 2: It is mentioned that it uses a spiking whisker sweep dataset in page 3 but it would be better if it is mentioned in the table descriptions as well for better clarity.
>
> **A7:** In the original manuscript, traditional ANNs utilize the raw real-valued whisker sweep dataset, while SNNs employ the spiking whisker sweep dataset introduced in our paper. We have added clarifications in Tables 1 and 2 to distinctly differentiate between the two. The supplementary experiments described in response A1 have also been included in Table 1.

---

> > ### Comment · Reviewer_uNoP · 2024-11-28
> >
> > Authors have addressed my concerns so I have revised the score accordingly.

---

### Official Review · Reviewer_YywY · 2024-11-04

**Soundness:** 3
**Presentation:** 3
**Contribution:** 3
**Rating:** 8
**Confidence:** 3

**Summary:**

The authors presents a biologically constrained model of the mouse barrel cortex, comprising 4,218 neurons across 13 neuronal subtypes, with neural distribution and connection strengths constrained by anatomical experimental findings.

The authors claim to develop an efficient training algorithm tailored for this model and convert an existing simulated whisker sweep dataset into a spiking-based format.

Their model achieves a classification accuracy exceeding classical ANN architectures by an average of 8.6%, and is on par with recent spiking neural networks (SNNs) in performance, and outperforms SNNs in whisker deprivation experiments.

Post-training analysis reveals that neurons within the model exhibit firing characteristics and distribution patterns similar to those observed in the actual neuronal systems of the barrel cortex.

**Strengths:**

-The authors develop an efficient training framework specifically designed for a biologically constrained model of the barrel cortex, aimed at achieving realistic sensory-motor integration.
-The authors meticulously construct a biologically realistic network comprising 4,218 neurons across 13 neuronal subtypes, ensuring an authentic replication of the biological structure of the barrel cortex.
-The classification accuracy of their model surpasses that of mainstream models like CNNs, RNNs, and LSTMs, and is competitive with recent SNNs, particularly in neuroscience-inspired whisker deprivation experiments.

**Weaknesses:**

- Reliability of the biological model: There are always free parameters difficult to be constrained, such as noise shape and exact synaptic efficacy. Small changes in these parameters can lead to completely different dynamical states and computational properties, and I am not sure the the CV is enough to asses the plausibility of the models activity. Can the author comment on how does this affect the generalizability of the findings?

- Generality of the comparison with standard ANNs: Why is this model so much better than ANN? Is the improvement due to spikes or to the architecture? How would an ANN perform if it had a similar topology inspired by the barrel cortex? Fig.6B makes me think that most of the improvement is due to the architecture. Can the author discuss on why?

**Questions:**

- Why can't the authors feed continuous signals into the spiking network and let it convert to spiking?
- In the comparison with convolutional networks, are they tested on the original dataset or on the spiking one?
- Occlusion results are interesting. A panel with performance decay across the different models would be very useful. What is the comparison with the mentioned biological experiments?
- Did the authors investigate the importance of the heterogeneity due to different neuron types? E.g. the comparison between homogeneous and uniform models classification performances. Also, I would ask the authors improve the introduction of dynamic gradients.

---

> ### Author Response · Authors · 2024-11-23
> **Response to Reviewer YywY (part 1)**
>
> Thank you for taking the time to review. Please see our detailed response to the issues you have raised.
> ## Response to Weaknesses
> > W1: Reliability of the biological model: There are always free parameters difficult to be constrained, such as noise shape and exact synaptic efficacy. Small changes in these parameters can lead to completely different dynamical states and computational properties, and I am not sure the the CV is enough to asses the plausibility of the models activity. Can the author comment on how does this affect the generalizability of the findings?
>
> **A1:** Thank you for inquiring about the reliability and generalizability of the Coefficient of Variation (CV).
> The sensitivity to parameters is a characteristic of biological neural networks and also the reason they are challenging to train. Therefore, we adopted the CV measure from studies on brain critical states ("Theoretical foundations of studying criticality in the brain," 2022) to quantify the complex internal states of our network. Its primary goal is to find an initial state in which a biological neural network can be trained.
>
> We believe **this approach is easily generalizable to other models constrained by biological circuits because of its simplicity and intuitiveness, indicating that no additional complex techniques are employed.** We hold the view that the more straightforward and intuitive a metric is, the better its reliability and generalizability. The **revised Figure 2B** presents a more comprehensive evolution of the CV, providing a visual glimpse into the network's state.
>
> Moreover, we consider that integrating findings from the analysis of brain dynamics into the training of biological neural networks will be a valuable direction for future research, including but not limited to the Kuramoto model and critical points (“Excitation–Inhibition Balance, Neural Criticality, and Activities in Neuronal Circuits,” 2024). Quantitative assessment of the roles played by various biological parameters and an in-depth analysis of dynamics, will be the focus of subsequent work.
>
> > W2: Generality of the comparison with standard ANNs: Why is this model so much better than ANN? Is the improvement due to spikes or to the architecture? How would an ANN perform if it had a similar topology inspired by the barrel cortex? Fig.6B makes me think that most of the improvement is due to the architecture. Can the author discuss on why?
>
> **A2:** Thank you for your concern about the different input formats for ANNs and SNNs. **We have supplemented our study with experiments where SNNs utilize real-valued inputs, and the results further confirm the performance superiority of our model.**
>
> The original manuscript indeed presented a discrepancy in that ANNs utilized real-valued inputs, while SNNs employed spiking-based inputs, which complicated the interpretation of results. Consequently, we have supplemented Table 1 with the results of SNNs tested on the original real-valued dataset. The content of Table 1 is also presented below:
>
> | ANNs | ST CNN | DB LSTM | RNN+ | UGRNN |
> |---|---|---|---|---|
> | Real-valued | 67.2 | 70.9 | 78.2 | 76.4 |
>
> | SNNs | SRNN 256 | STSC SNN | TA SNN | Ours |
> |---|---|---|---|---|
> | Spiking-based | **81.8** | 80.0 | **81.8** | **81.8** |
> | Real-valued | 85.5 | 87.3 | 83.6 | **89.1** |
>
> It can be observed that the performance of all SNNs improved on the real-valued dataset, with our barrel model maintaining an advantage. The enhancement in SNNs is understandable: as described in Section 3.3, random sampling of independent spikes to represent real values at each moment introduces perturbations. Additionally, expanding the original dataset's time step from 110 to 550 is also detrimental to the firing of spiking neurons.
>
> Besides, **using detailed anatomical structures to constrain ANNs presents two challenges:** **1.** How to model the diversity of neuronal subtypes within ANNs? **2.** How to depict the complex and variable circuitry structures within ANNs? Given that current ANNs employ uniform simple neuron models and regular connection patterns, this direction remains open for research.
>
> **As for the advantages of biologically plausible models, we believe they stem from their unique structures.** In fact, the brain's architecture has also been optimized through the process of biological evolution, encoded in genes ("Probabilistic skeletons endow brain-like neural networks with innate computing capabilities," 2021). Specifically, the diversity of neurons and their anatomical connections together form a stable network state, and disruptions to these connection probabilities destabilize this network state. Our work is dedicated to bridging the gap between biological neural networks and artificial intelligence to foster a deeper understanding of the functions of brain components.

---

> ### Author Response · Authors · 2024-11-23
> **Response to Reviewer YywY (part 2)**
>
> This is the continued response to the remaining questions you have posed.
> ## Response to Questions
> > Q1: Why can't the authors feed continuous signals into the spiking network and let it convert to spiking?
>
> **A3:** We utilize spiking-based inputs in accordance with the fundamental rule that biological neurons in the brain transmit signals through spikes, thereby enhancing the biological plausibility of our model. Technically, real-valued data can also be used to train SNNs, and we have supplemented our experiments as detailed in our response to W2.
>
> > Q2: In the comparison with convolutional networks, are they tested on the original dataset or on the spiking one?
>
> **A4:** In the original manuscript, all ANNs were trained using the original real-valued dataset (Zhuang et al., 2017), while all SNNs were trained using the spiking-based dataset that we converted. As detailed in our response to W2, we have followed the suggestion and supplemented the experimental results of SNNs on the real-valued dataset.
>
> >Q3: Occlusion results are interesting. A panel with performance decay across the different models would be very useful. What is the comparison with the mentioned biological experiments?
>
> **A5:** Following the comment, we provided a figure depicting the performance degradation of whisker deprivation experiments as detailed in Section 4.1, which aligns with the results in Table 2, showing that our barrel model exhibits the slowest performance degradation (**see Figure 2 in the supplementary material**).
>
> >Q4: Did the authors investigate the importance of the heterogeneity due to different neuron types? E.g. the comparison between homogeneous and uniform models classification performances. Also, I would ask the authors improve the introduction of dynamic gradients.
>
> **A6:**  Yes, we had previously considered constructing a model devoid of neuronal diversity for comparative purposes. However, due to our method's constraint that all synaptic weights be positive (see in Equation 4), this model was nearly impossible to train effectively. This may be related to the pre-established excitatory-inhibitory balance inherent in brain networks.
>
> To provide a more comprehensive response to your question, **we additionally constructed two models, an SRNN and an SFNN, that are identical in scale and posses the same read-in and read-out pathways as our model for comparative analysis.** The results are shown in the following table：
>
> |Ours|SFNN 4418|SRNN 4418|
> |---|---|---|
> |**81.8**|74.6|48.8|
>
> The supplementary results indicate that the performance of our model is not attributable to the scale of its neuronal population. This experiment has been included **in Figure 8 of the supplementary material**.
>
> Moreover, following your suggestion, we have refined the content about dynamic gradients in Section 4.3 to make it clearer.

---

> > ### Comment · Reviewer_YywY · 2024-11-26
> >
> > Authors have addressed my concern. I will raise my score.

---

### Author Response · Authors · 2024-11-26
**Summary of Major Modifications in Our Manuscript**

We sincerely appreciate the professional and constructive comments from all reviewers. In light of these comments, we have conducted supplementary experiments and revised our manuscript, which significantly improved our work. Below is a summary of the **major modifications** in our manuscript:

**1. Manuscript Title:** We have revised the title to 'Effective Integration of Whisker Sensory Information and Replication of Key Biological Network Statistics by a Biologically Constrained Model of the Mouse Barrel Cortex' to better align with our contributions.

**2. Comparison on Real-valued Datasets:** We have conducted additional tests on the performance of SNNs on the real-valued whisker sweep dataset. The results further confirm the advantages of our barrel cortex model and have been added to Table 1.

**3. Comparison of Firing Selectivity in Other Models:** We performed additional analyses of the firing selectivity of the remaining three SNNs using the same method in Section 4.2. The results show that significant firing selectivity is a unique feature of our model, and these findings have been added to Figure 3 in the supplementary material.

**4. Improvement of Writing in Section 4.3:** We have re-articulated the experiment on the similarity of neuronal subtype dynamics to make it clearer. We also clarified the calculation process and purpose of normalization, emphasizing that it does not affect the computation of Pearson correlation coefficient.

**5. Comparison of Degree Distribution in Other Models:** We performed additional statistics on the degree distribution of SRNN 256 and LSTM models using the same method described in Section 4.4. The results show that a significant long-tail effect is a unique feature of our trained model, and these findings have been added to Figure 6 in the supplementary material.

**6. Detailed Display of Neuronal Activity:** We presented the raster plots of the pre- and post-training model in Figure 1 of the supplementary material, with the firing rates of each population.

**Below, we provide detailed responses to the comments from each reviewer.**

---

### Meta-Review · Area_Chair_1MLz · 2024-12-16

**Metareview:**

This paper presents a model of rodent barrel cortex, consisting of thousands of integrate-and-fire neurons with 13 distinct neuronal subtypes, and anatomical connections constrained by empirical observations. As well, the authors develop a technique to fit this model to behavioural data by introducing an order parameter to measure the synchrony of neuronal activity, and using this to identify trainable initial states. The authors claim that their model is superior to other models in terms of its ability to learn to classify objects using simulated whisker touch, and that their model can predict the outcome of whisker deprivation experiments.

The strengths of this paper are that it clearly backs up its claims of their own model's performance with empirical evidence, and helps to improve the biological plausbility of models of behaviorally relevant models of barrel cortex. The weaknesses are that the controls for model comparisons were not necessarily fair comparisons, and that the actual novelty of the training approach is limited. However, based on the discussions and final scores, a decision of accept (poster) was reached.

**Additional Comments On Reviewer Discussion:**

The discussion was robust, with most reviewers engaging the authors. The authors worked hard to address the concerns, and in the end, the final scores were all at or above the acceptance threshold (6).

---

### Decision · Program_Chairs · 2025-01-22

Accept (Spotlight)